# Requirement of a functional ion channel for Sindbis virus glycoprotein transport, CPV-II formation, and efficient virus budding

Zeinab Elmasri[1], Vashi Negi[2], Richard J. Kuhn[2,3], Joyce Jose[1,4]*

1 Department of Biochemistry and Molecular Biology, The Pennsylvania State University, University Park, Pennsylvania, United States of America, 2 Department of Biological Sciences, Purdue University, West Lafayette, Indiana, United States of America, 3 Markey Center for Structural Biology and Purdue Institute of Inflammation, Immunology and Infectious Disease, Purdue University, West Lafayette, Indiana, United States of America, 4 The Huck Institutes of the Life Sciences, The Pennsylvania State University, University Park, Pennsylvania, United States of America

* jxj321@psu.edu

**Data Availability Statement:** All relevant data are within the paper and its Supporting Information files.

## Abstract

Many viruses encode ion channel proteins that oligomerize to form hydrophilic pores in membranes of virus-infected cells and the viral membrane in some enveloped viruses. Alphavirus 6K, human immunodeficiency virus type 1 Vpu (HIV-Vpu), influenza A virus M2 (IAV-M2), and hepatitis C virus P7 (HCV-P7) are transmembrane ion channel proteins that play essential roles in virus assembly, budding, and entry. While the oligomeric structures and mechanisms of ion channel activity are well-established for M2 and P7, these remain unknown for 6K. Here we investigated the functional role of the ion channel activity of 6K in alphavirus assembly by utilizing a series of Sindbis virus (SINV) ion channel chimeras expressing the ion channel helix from Vpu or M2 or substituting the entire 6K protein with full-length P7, in cis. We demonstrate that the Vpu helix efficiently complements 6K, whereas M2 and P7 are less efficient. Our results indicate that while SINV is primarily insensitive to the M2 ion channel inhibitor amantadine, the Vpu inhibitor 5-N, N-Hexamethylene amiloride (HMA), significantly reduces SINV release, suggesting that the ion channel activity of 6K similar to Vpu, promotes virus budding. Using live-cell imaging of SINV with a mini-iSOG-tagged 6K and mCherry-tagged E2, we further demonstrate that 6K and E2 colocalize with the Golgi apparatus in the secretory pathway. To contextualize the localization of 6K in the Golgi, we analyzed cells infected with SINV and SINV-ion channel chimeras using transmission electron microscopy. Our results provide evidence for the first time for the functional role of 6K in type II cytopathic vacuoles (CPV-II) formation. We demonstrate that in the absence of 6K, CPV-II, which originates from the Golgi apparatus, is not detected in infected cells, with a concomitant reduction in the glycoprotein transport to the plasma membrane. Substituting a functional ion channel, M2 or Vpu localizing to Golgi, restores CPV-II production, whereas P7, retained in the ER, is inadequate to induce CPV-II formation. Altogether our results indicate that ion channel activity of 6K is required for the formation of CPV-II from the Golgi apparatus, promoting glycoprotein spike transport to the plasma membrane and efficient virus budding.

**Funding:** This work was supported by NIH grants GM056279 and AI095366 to R.J.K. and The Pennsylvania State University Startup funds to JJ. The funders had no role in study design, data collection and analysis, decision to publish, or preparation of the manuscript.

**Competing interests:** The authors have declared that no competing interests exist.

## Author summary

Alphaviruses cause acute and long-term diseases ranging from febrile illness to fatal encephalitis in infected individuals. Identifying new drug targets for antiviral development is crucial to controlling and combating alphavirus infections. Inhibitors for viral ion channels such as Influenza M2, HIV Vpu, and HCV P7 have been proven to be effective antivirals. Alphavirus 6K is a 6 kDa structural protein that forms ion channels on membranes of infected cells. However, no structural information is available to establish its functional role in alphavirus life cycle. Using a miniSOG-tagged 6K SINV, we determined the spatial and temporal organization of 6K for the first time in infected cells. Our results reveal the colocalization of 6K with E2 in the Golgi, where its ion channel activity is essential for CPV-II formation. This study establishes that 6K is required for efficient glycoprotein transport to the plasma membrane, a function that can be partially complemented in cis by other viral ion channels that traffic to the Golgi. We demonstrate that the inhibition of 6K by a channel-blocking drug, 5-N, N-Hexamethylene amiloride impedes alphavirus budding, laying the foundation for developing inhibitors targeting 6K as an attractive antiviral strategy.

## Introduction

Sindbis virus (SINV) is an enveloped, mosquito-borne alphavirus belonging to the family *Togaviridae*. Alphaviruses cause debilitating acute polyarthritis and encephalitis in humans and pose a significant risk to global health due to their wide geographic distribution and their ability to cause severe diseases in humans and other animals [1,2]. Reemerging alphaviruses such as chikungunya virus (CHIKV) and Mayaro virus (MAYV) have become a significant health problem in Central and South America, causing debilitating long-term arthritis [3–6]. Venezuelan equine encephalitis virus (VEEV) and Eastern equine encephalitis virus (EEEV) can cause severe and sometimes fatal encephalitis in humans [7,8]. Currently, there are no treatments or vaccines licensed for use against alphavirus infections.

Alphavirus virions are spherical with a diameter of approximately 70 nm. The 11.7 Kb positive-sense single-stranded RNA genome is packaged into a nucleocapsid core (NC) surrounded by a host-derived lipid bilayer [9]. The 5' two-thirds of the genome encodes four non-structural proteins (nsP1-4) that, along with host proteins and viral RNA, form the membrane-associated alphavirus replication complex (RC) [10,11]. The 3' one-third of the genome encodes the structural proteins capsid (CP), E3, E2, 6K, Trans frame (TF), and E1 that are translated from a subgenomic RNA. Trimers of the E1/E2 heterodimers form 80 trimeric spikes on the surface of the virion [1]. Alphavirus budding in mammalian cells occurs at the plasma membrane following specific interactions of the NC and the cytoplasmic domain of E2 (cdE2) [12–14]. In mosquito cells, virus budding occurs at the plasma membrane as well as at the membranes of large cytoplasmic vesicles [15]. Electron microscopy (EM) analyses of alphavirus-infected mammalian cells have revealed two types of virus-induced structures called cytopathic vacuoles type 1 (CPV-Is) and type II (CPV-IIs) [16]. CPV-Is are larger and associated with replication complexes (RC) [15,17,18]. CPV-IIs are assembly structures with a diameter of 0.5–1 μm and are derived from the *trans*-Golgi network [19,20]. Electron tomography studies have shown that NCs are attached to the cytoplasmic side of CPV-IIs. E1 and E2 are found within these vacuoles arranged in hexagonal arrays similar to their arrangement on the virion envelope [21]. The near-atomic structure of the CHIKV RNA replicase and the

subnanometer-resolution cellular architecture of the RCs have been resolved recently utilizing *in vitro* reconstitution and *in situ* electron cryotomography [22]. Similarly, recent innovative TEM techniques have revealed the presence of four different morphological classes of CPV-IIs, with class 1 originating directly from swelling of the Golgi, and classes 2, 3, and 4 originating from fragmentation and bending of the Golgi cisternae [20]. However, the precise mechanisms through which NCs attach to the CPV-II and how NCs arrive at the plasma membrane are yet to be determined. The clustering of CPV-IIs near the plasma membrane suggests that these vacuoles might play a role in delivering NCs to the plasma membrane for budding.

The structures of several alphaviruses such as SINV [23,24], Semliki forest virus (SFV) [25], EEEV [26], and more recently MAYV and VEEV [27,28], were resolved using cryo-electron microscopy (cryo-EM) and have provided considerable knowledge about the functions of E1 and E2 in the alphavirus life cycle. The 6K and TF have been reported to be incorporated into the virions in substoichiometric amounts [29,30]. The recent 3.0 Å resolution VEEV virus-like particles (VLPs) [28] and high-resolution cryo-EM maps of alphaviruses have not revealed the density corresponding to either 6K or TF. The alphavirus 6K is a small hydrophobic protein predicted to have two transmembrane helices, with the second helix serving as a translocation signal for E1 [31]. During translation, the presence of a slippery codon motif in the 6K gene can result in ribosomal frameshifting leading to the production of TF while hindering E1 translation [32]. Although 6K and TF have the same N-terminal transmembrane helix, the two proteins differ in that TF has a unique cytoplasmic C-terminal domain which is conserved among alphaviruses, while 6K has a second transmembrane helix [32]. In the context of the SFV life cycle, the role of 6K has been shown to be dispensable for virus entry, replication, and glycoprotein trafficking to the plasma membrane, but is essential for virus budding [33,34]. However, a partial deletion of SINV 6K and a complete deletion of Salmonid alphavirus (SAV) 6K both have resulted in aberrant glycoprotein processing and trafficking in addition to budding defects [35,36].

Bacterial expression of 6K protein has been found to alter *E. coli* membrane permeability, a characteristic that primarily led to the classification of 6K as a viral ion channel or viroporin [37–39]. Furthermore, mutations of the interfacial domains of 6K have indicated the presence of an oligomeric channel that selectively transports divalent cations [40]. Viral ion channels play essential roles in the life cycles of RNA viruses [41,42]. Influenza A virus (IAV) M2 ion channel is a proton channel that regulates the pH during virus entry and glycoprotein trafficking during exit [43,44]. Human immunodeficiency virus type 1 (HIV-1) Vpu promotes virus budding while also important for counteracting human tetherin [45–47]. Hepatitis C (HCV) P7 orchestrates virus assembly and regulates the pH maturation process of HCV particles [48,49]. The key roles played by viral ion channels also make them attractive therapeutic targets. Amantadine and amiloride analogs that inhibit ion channels have showed efficacy as antiviral drugs against multiple viruses such as IAV, HIV-1, and HCV [50–52]. Previously, Vpu expression *in trans* has been shown to complement the membrane permeabilization capacity and virus particle production of SINV 6K with a partial deletion [53] while also suggesting that 6K is functionally analogous to Vpu.

In this study, we demonstrate the functional role of the ion channel activity of 6K in alphavirus glycoprotein trafficking and budding. By incorporating a miniSOG tag [54], we determined the real-time localization of 6K for the first time by live imaging in SINV infected cells. We show that 6K and E2 colocalize on Golgi in the secretory pathway, which is unaffected by the deletion of TF. We observed a similar 6K and E2 colocalization in virus-infected mosquito cells. By using SINV with 6K deletion (SINV Δ6K) we found that 6K is critical for biogenesis of CPV-IIs and efficient glycoprotein trafficking to the plasma membrane. To investigate whether these defects are due to the absence of a functional ion channel, we first generated

recombinant SINV expressing full-length P7 in place of 6K. Although the full-length P7 substitution improved the rate of virus budding compared to the 6K deletion, it was still defective compared to wild-type virus and failed to rescue the CPV-II formation. We reasoned that the defect is due to the site of action as HCV assembly and budding occur in association with the ER membrane, unlike alphaviruses that bud from the plasma membrane [55]. Using Flag-tagged 6K and P7 viruses, we compared the subcellular localization of P7 with that of 6K and found that in the context of SINV chimera, the P7 is retained in the ER, whereas 6K localizes to ER and Golgi. Subsequently, we generated chimeric SINV using ion channels M2 and Vpu encoded by enveloped viruses IAV and HIV-1 respectively, that bud from the plasma membrane. We constructed these chimeras by substituting the ion channel transmembrane helix of 6K with that of HIV-1 Vpu, and IAV M2. Since M2 and Vpu are Class I viroporins with a single membrane-spanning domain [42], to ensure the correct membrane topology of the structural polyprotein, we retained the C-terminal helix of 6K in both M2 and Vpu chimera. Confocal imaging and EM analysis of cells infected with the chimeric M2 SINV and Vpu SINV show that glycoprotein trafficking and CPV-II formation are comparable to that of WT SINV. Immunostaining analysis of cells infected with the Flag-tagged chimeric viruses indicate that 6K, M2, and Vpu localize to the Golgi apparatus, unlike P7, which is retained in the ER. Our results demonstrate that CPV-II formation and efficient glycoprotein trafficking require the ion channel activity of 6K in the late secretory pathway since they were complemented by Vpu and M2. Our study attributes specific functions to 6K in regulating virus assembly, which can be inhibited by channel blockers, thus presenting 6K as an antiviral target for alphavirus drug development.

## Materials and methods

### Viruses and cells

BHK-15 (baby hamster kidney) cells were maintained at 37˚C and 5% $CO_2$ in Eagle's minimum essential medium (ThermoFisher) supplemented with 10% fetal bovine serum (FBS, Seradigm #1500–500), nonessential amino acids (Gibco, #11140–050), 1X Penicillin-Streptomycin solution (Corning Inc). C6/36 mosquito (*Aedes albopictus*) cells (ATCC) were maintained at 30˚C in the presence of 5% $CO_2$ in MEM supplemented with 2 mM L-glutamine, 0.1 mM nonessential amino acids, 25 mM HEPES, and 10% heat-inactivated FBS. Huh-7.5 (human hepatoma) cells were maintained at 37˚C, and 5% $CO_2$ in Dulbecco's modified Eagle's medium (ThermoFisher) supplemented with 10% FBS, 1X Penicillin-Streptomycin solution, L-glutamine, and nonessential amino acids. Wild type and mutant SINV were generated from a full-length cDNA clone pToto64 [56].

### Plasmids and cloning

Alphavirus 6K sequence were analyzed and multiple sequence alignment were generated using Clustal omega [57]. Alignments were viewed using Jalview [58]. (S1 Fig). The 6K mutations were generated in pToto64 by standard overlap PCR mutagenesis procedures. Overlap PCR products were digested with *Bss*HII and *Bsi*WI and cloned into pToto64. Sequences corresponding to influenza M2 transmembrane helix, HIV Vpu transmembrane helix, and full-length HCV-P7 were synthesized as oligonucleotide primers for overlap extension PCR (S1 Table). A FLAG (DYKDDDDK) epitope tag was inserted after the N-terminal signalase cleavage site on 6K and other ion channel chimeras by overlap PCR. Sequence corresponding to the fluorescent protein miniSOG [54] was amplified and cloned at the N-terminus of 6K after the signalase cleavage site by overlap PCR. Deletion of 6K was also introduced into a previously described fluorescent protein (FP) tagged SINV construct where the mCherry was inserted as

an N-terminal tag of E2 after the E3-E2 furin cleavage site [59]. The dual-labeled miniSOG-6K/mCherry-E2 construct was generated by subcloning a *Bss*HII-*Bsi*WI insert from the miniSOG-6K plasmid into the mCherry-E2 plasmid. ΔTF construct was generated using primers (S1 Table) as previously described [37]. For the construction of mammalian expression plasmids, the mCherry-tagged structural polyproteins E3-mCherry-E2-6K-E1, CP-E3-mCherry-E2-6K-E1, and E3-mCherry-E2-E1 were amplified from full-length SINV cDNA clones of mCherry-E2 and Δ6K mCherry-E2 by PCR and subcloned into pcDNA3.1 plasmid for mammalian expression. S1 Table lists all the oligonucleotide primers used in this study.

### *In vitro* transcription and transfection

Full-length infectious RNA of the wild-type (WT) and mutant SINV were generated by in vitro transcription using the WT and mutant pToto64 cDNA clones. The cDNA clones were linearized with *Sac*I followed by *in vitro* transcription with SP6 RNA polymerase as previously described [60]. BHK-15 cells were electroporated with 10 μg of *in vitro* transcribed RNA. Virus-containing media were harvested at 24 hours post-electroporation and the virus titers (plaque-forming units per ml) were determined by titration on BHK-15 cells monolayers. The presence of mutation in each virus was confirmed by sequencing the reverse transcription (RT)-PCR products corresponding to the 6K coding region from RNA purified from the cytoplasmic extracts of infected BHK-15 cells.

### Plaque assay

Plaque assays were performed on BHK cells, and the number of plaques was determined after staining the plaques with either neutral red or crystal violet. For neutral red staining, virus stocks were serially diluted in PBS supplemented with 1% FBS and 1 mM each of $CaCl_2$ and $MgCl_2$. From the virus stocks, 250 μL were added to each well of BHK-15 cell monolayers grown on six-well plates and incubated at room temperature for 1 h with rocking. The cells were subsequently overlaid with 3 ml of 1% agarose in MEM and incubated at 37°C with 5% $CO_2$. After 48 hours, virus titers were determined by staining the plates with neutral red (MilliporeSigma, #N2889) and counting the number of plaques. For crystal violet staining, serially diluted virus stocks were added to each well of a BHK-15 monolayer of cells grown on a 24-well plate and rocked for 1 hour at room temperature. Subsequently, the cells were overlaid with MEM containing 2% FBS, 2% cellulose, and 2% 20 mM HEPES buffer pH: 7.4. Plates were incubated for 48 h at 37°C in the presence of 5% $CO_2$. Plaques were counted after fixing the cells for 2 h with a mixture of 10% formaldehyde and 2% methanol (v/v in water) and staining with 0.1% crystal violet prepared in 20% ethanol. All experiments were performed in triplicates.

### Growth kinetic analyses

Growth kinetics of wild-type and mutant SINV were determined by one-step growth curve analysis as previously described [15]. Briefly, BHK-15 cells on 6 well plates were infected with WT and mutant viruses at a multiplicity of infection (MOI) of 2 and rocked for 1 hour at room temperature. After infecting the cells for 1 hour at 37°C virus inoculum was removed, and cells were washed twice with PBS supplemented with 1% FBS and 1 mM each of $CaCl_2$ and $MgCl_2$. Cells were then incubated at 37°C in MEM supplemented with 5% FBS. Supernatants were collected from infected cells and the medium over cells was replaced with fresh medium every 1 hour for 12 hours post-infection (hpi). Virus titers at different time points post-infection were determined by plaque assays using a monolayer of BHK-15 cells. All experiments were performed in triplicates.

## qRT-PCR and specific infectivity

A quantitative real-time PCR assay was performed to determine the number of virus particles released at different time points post-infection, as previously described [15]. Briefly, RNA was extracted from the culture supernatant using RNeasy kit (Qiagen, Valencia, CA, USA) according to the manufacturer's instructions. SuperScript III Platinum SYBR green one-step qRT-PCR kit (Invitrogen, Grand Island, NY) was used to perform qRT-PCR in triplicate in 25-μl sample volumes that contained a 5-μl aliquot of purified viral RNA. SINV-specific primers 5′ TTCCCTGTGTGCACGTACAT 3′ and 5′ TGAGCCCAACCAGAAGTTTT 3′, which bind nucleotides 1044 to 1063 and nucleotides 1130 to 1149 of the SINV genome, respectively were used for the assay. Experiments were performed in triplicate. The standard curve of the cycle threshold ($C_T$) used to determine the viral RNA copy number was generated using *in vitro* transcribed genomic RNA. The cycling parameters were 4 min at 50°C and 5 min at 95°C, followed by 40 cycles of 5 s at 95°C and 1 min at 60°C. All experiments were performed in triplicates.

Specific infectivity of WT and chimeric viruses was determined by calculating the ratio of the average number of viral RNA molecules per milliliter divided by the average number of PFU per milliliter as previously described [61–63]

## Thin-section TEM

BHK-15 cells infected with WT or mutant SINV at an MOI of 2 were fixed at 6- or 12-hours post-infection. Samples were fixed with 2% glutaraldehyde–0.1 M cacodylate buffer (0.1 M Na-cacodylate, 2 mM $MgCl_2$, 2 mM $CaCl_2$, and 0.5% NaCl at pH 7.4) for 3 days, washed in cacodylate buffer followed by water, embedded in 2% agarose, postfixed for 90 min in buffered 1% osmium tetroxide containing 0.8% potassium ferricyanide, and stained for 45 min in 2% uranyl acetate. Following that, the samples were dehydrated with a graded series of ethanol, embedded in EMbed-812 resin, and cell sections were cut using Reichert-Jung Ultracut E ultramicrotome. Images of samples stained with uranyl acetate and lead citrate were then obtained in an FEI Tecnai $G^2$ 20 electron microscope equipped with a $LaB_6$ source and operated at 100 keV (Life Science Microscopy Facility, Purdue University). A minimum of 50 frames were collected for each condition.

## Immunofluorescence (IF) Analysis

IF was performed after growing BHK-15 cells or Huh-7.5 cells on glass coverslips. Briefly, cells were either transfected with mEmerald-SEC61B-C1 (Addgene plasmid# 90992) and infected with Flag-6K SINV, or Flag-tagged chimeric SINV, or infected with WT SINV, Δ6K SINV, or Chimeric SINV. Cells were then fixed for 15 min at room temperature with 4% paraformaldehyde in phosphate-buffered saline (PBS) at 6 or 12 hpi. The fixed cells were permeabilized with 0.1% Triton-X100 in PBS for 5 min. a mouse monoclonal anti-E2 antibody and a rabbit polyclonal anti-CP antibody were used to detect E2 and CP. A mouse monoclonal anti-FLAG antibody (Sigma) was used to detect the FLAG tag. A rabbit polyclonal anti-Giantin antibody (Abcam, ab80864) was used to image the Golgi apparatus. The secondary antibodies used were fluorescein isothiocyanate (FITC) or tetramethylrhodamine isothiocyanate (TRITC)-conjugated goat anti-rabbit and goat anti-mouse antibodies (Thermo Fisher Scientific). Nuclei were stained with Hoechst stain (Invitrogen), and images were acquired using a Nikon AIR confocal microscope with a 60X oil objective. The NIS Elements software (Nikon) was used for processing images.

## Western blot analysis

BHK-15 cells were infected with P2 (second passage) stock of viruses recovered from cells electroporated with in-vitro transcribed RNA of wild-type SINV, Δ6K SINV, and miniSOG-6K SINV. Infected cells were lysed using a lysis buffer [25mM Tris-HCl pH 7.5, 150 mM NaCl, 1% Triton-X 100, 5mM 2-mercaptoethanol, 1mM PMSF] at 12 hpi. Lysates were separated on a 4–20% precast SDS-PAGE gel (Bio-Rad). Proteins were transferred onto a nitrocellulose membrane and probed with mouse monoclonal anti-actin antibody and rabbit polyclonal antibodies against SINV E2 and Capsid proteins. Infrared labeled secondary antibodies (goat anti-mouse 680 and goat anti-rabbit 800) were used for detection. Blots were imaged using Odyssey CLx Near-Infrared Imaging System and analyzed using Licor Image Studio software.

## Flow cytometry analysis

Flow cytometry analysis was performed as previously described [64]. BHK-15 cells were seeded on 35mm dishes and infected with WT or mutant SINV with an MOI of 2. Infected cells were then trypsinized at 6, or 12 hpi and resuspended with MEM containing 5% FBS. Cells were incubated for an hour on ice with a 1 to 50 dilution of a monoclonal mouse anti-E2 antibody and subsequently washed 3 times with PBS. A goat anti-mouse FITC secondary antibody was then used to stain the cells. Flow analysis was performed on a Beckman Coulter FC500 flow cytometer and with the FlowJo software package. Experiments were performed in duplicates.

## SINV infection and drug treatment

BHK-15 cells seeded in a 96-well plate were treated with different concentrations of Amantadine or HMA for 12 hours. Cells were then infected with the viruses at an MOI of 0.1 and incubated at 37°C in 5% $CO_2$. Supernatants were collected at 8 hpi and virus titers were determined using plaque assay conducted on a monolayer of BHK-15 cells followed by crystal violet staining. Experiments were performed in triplicates.

## Cytotoxicity assay

BHK-15 cells were seeded into 96-well plates (Thermo Fisher Scientific) and treated with the indicated different concentrations of Amantadine or Hexamethylene amiloride (HMA) after 24 hours. Cells were then incubated with compounds at 37°C in 5% $CO_2$ for 24 hours. For HMA, DMSO-treated cells were used as controls. Cytotoxicity was determined using alamarBlue cell viability reagent (BIO-RAD) according to the manufacturer's instructions. In short, alamarBlue was added to the 96-well plates in an amount equal to 10% of the volume in the well. The plates were then incubated at 37°C in 5% $CO_2$ for 3 hours. Absorbance at 570 nm and 600 nm was determined using an ELISA plate reader (SpectraMax iD3). Data analysis was performed using PRISM 9 software.

## Live-cell imaging

BHK-15 cells seeded in a 4-well chamber (Ibidi) were infected with 6K-miniSOG or mCherry-E2 virus and imaged. For imaging using cellular markers, BHK-15 cells were transfected with the fluorescent-protein tagged mammalian expression plasmid YFP-membrane [65] using Lipofectamine 2000 (Thermo Fisher Scientific) according to the manufacturer's instructions. The mammalian expression constructs of E3-mCherry-E2-6K-E1 and E3-mCherry-E2-E1 were transfected using lipofectamine and subjected to imaging. For the live-cell imaging of fluorescent-protein tagged SINV, BHK-15 cells were electroporated with the *in vitro* transcribed RNA corresponding to the mCherry-E2 SINV and Δ6K-mCherry-E2 SINV and plated

on a 4-well chamber and imaged at indicated time-points. For detecting the subcellular localization of 6K, transfected cells expressing the fluorescent protein-tagged cellular markers mCherry-SEC61B-C1 (Addgene plasmid# 90994) or mTagBFP2-SiT-N-15 (Addgene plasmid# 55325) and infected with miniSOG-6K at an MOI of 5 and imaged at 12 hpi. Nikon AIR confocal microscope was used for live imaging using a growth chamber (Tokai Hit, Fujinomiya, Shizuoka Prefecture, Japan) supplied with 5% $CO_2$ at 37˚C using a 60X oil objective with 1.4 numerical aperture (NA). A Resonance scanner was used for miniSOG imaging. The lasers and emission band-passes used for imaging were as follows: blue, excitation of 405 nm and emission of 425 to 475 nm; green, excitation of 488 nm and emission of 500 to 550 nm; red, excitation of 561 nm and emission of 570 to 620 nm.

## Results

### 6K exerts its functional role during alphavirus release that in part can be complemented in cis by ion channels from enveloped viruses

To assess the role of 6K in Sindbis virus assembly, we constructed a SINV with complete 6K deletion (Δ6K SINV) (Fig 1A). The Δ6K SINV exhibited a small plaque phenotype compared

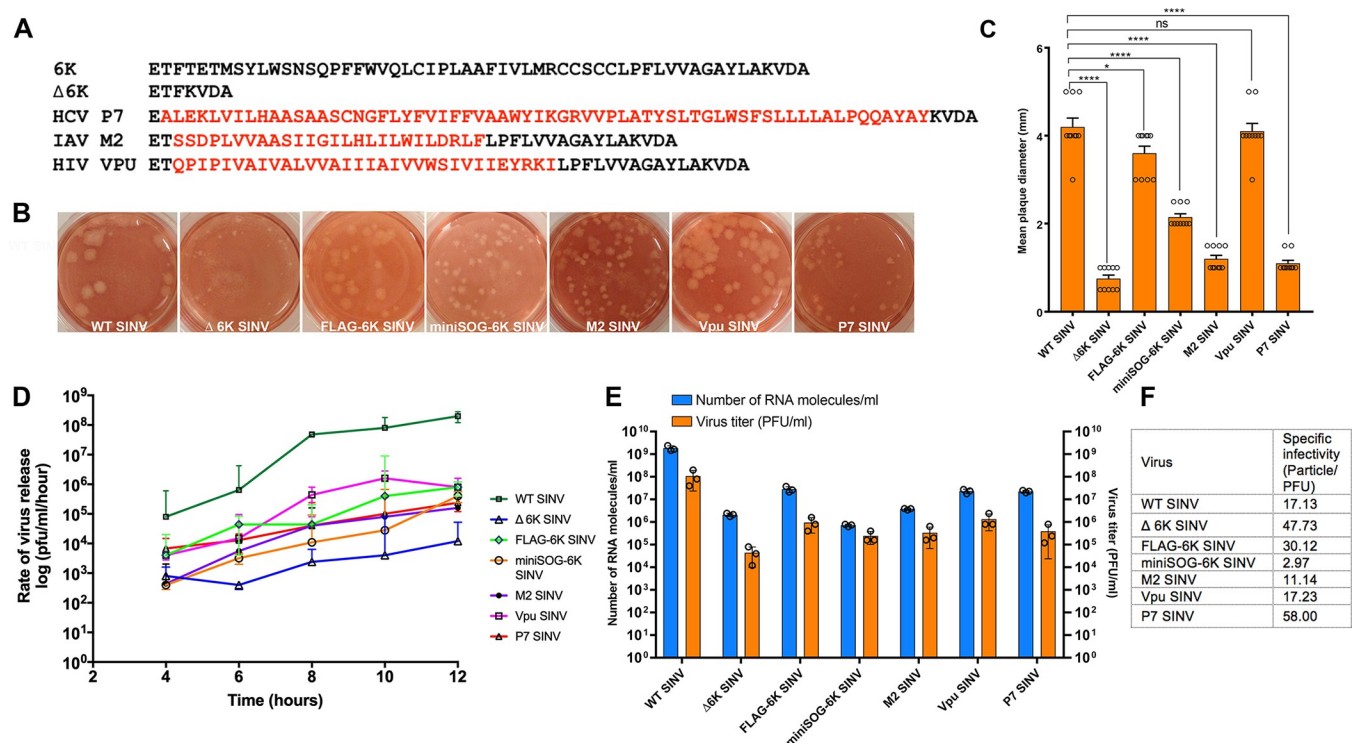

**Fig 1. Generation and characterization of 6K mutants. (A).** The amino-acid sequence of the 6K region from WT SINV, Δ6K SINV, p7 SINV, M2 SINV, and Vpu SINV. **(B).** Plaque morphologies of WT and 6K mutant viruses. Viruses were collected from electroporated BHK-15 cells 24 h post-electroporation, and plaques were visualized by staining BHK-15 cells 48 hpi. **(C).** Mean plaque diameters of the virus plaques in BHK-15 cells measured on day 2 pi. Plaque size was determined by random selection of 10 plaques and values are expressed as the mean with standard error (SEM). Significance was determined using Dunnett's multiple comparisons test as part of one-way ANOVA with a 95% confidence interval. *p* Values were considered significant when $p < 0.05$ (*), $p < 0.01$ (**), $p < 0.001$(***), and $p < 0.0001$(****). ns indicates "not significant". **(D).** One-step growth curve analysis of WT and 6K mutant viruses from BHK-15 cells. BHK-15 cells were infected with WT or 6K mutant viruses at an MOI of 5, medium was harvested and replaced every hour for 12 h, and the rate of virus release (plaque-forming unit (pfu) per ml per hour) was determined using standard plaque assays. Data shown are from three independent experiments. Error bars indicate standard error of mean (SEM). **(E).** Quantification of the number of virus particles released into the medium for WT and 6K mutant viruses at 12 hpi. The total number of viral genomic RNA molecules was determined by qRT-PCR using a standard curve of a known amount of in vitro-transcribed SINV RNA molecules. The pfu in these samples were determined by standard plaque assays of the virus supernatant collected at 12 hpi from infected BHK-15 cells. Data shown are from three independent experiments. The error bars indicate the standard error of mean (SEM). **(F)** Specific infectivity of WT and chimeric viruses.

to the large plaque phenotype of the WT SINV (Fig 1B and 1C). We next generated IAV M2 and HIV-1 Vpu ion channel chimeras to investigate whether the ion channel activity from other viral ion channels could functionally complement the ion channel activity of 6K in cis (Fig 1A). We generated these ion channel chimeras by substituting the conserved 6K transmembrane (TM) helix (S1 Fig) with ion channel TM helix from HIV-1 Vpu, and IAV M2 transmembrane helix, in cis. In the TM helix chimeras, the C-terminal signal sequence of 6K was retained for maintaining the polyprotein topology required for the precise E1 translocation. The slippery codon responsible for the translation of TF protein was removed from all the 6K mutants to inhibit TF production. Since HCV P7 and 6K have the same membrane topology as Class II viroporins with two transmembrane helices that span the membrane with their N and C termini in the ER lumen, we generated a full-length P7 chimera substituting the 6K sequence, except for the N and C terminal signalase cleavage sites (Fig 1A). Chimeric SINV produced plaques with morphology ranging from small to large (Fig 1B and 1C). We next assessed the one-step growth kinetics of WT and mutant SINV. We found that the deletion of 6K has a significant effect on virus release as it resulted in an approximately 4-log reduction in virus titer compared to WT SINV (Fig 1D). Vpu SINV exhibited a two-log increase in titer compared to Δ6K SINV followed by M2 SINV and P7 SINV (~1 log) (Fig 1D). Our results from P7 SINV indicate that a full-length viral ion channel with a similar membrane topology can partially substitute for 6K in cis. To ascertain whether these reductions in virus titers of the mutants are due to the presence of several noninfectious particles released into the media, we quantified the number of viral RNA molecules released into the media using qRT-PCR, and the particle/PFU ratio was calculated at the indicated time points (Fig 1E and 1F). The specific infectivity of the Δ6K SINV was reduced by ~2.8 folds compared to that of WT SINV. Among the chimeric viruses, the specific infectivity of P7 SINV was reduced by ~3.4 folds compared to WT SINV. On the other hand, M2 SINV and Vpu SINV exhibited specific infectivity comparable to the WT (Fig 1F) suggesting that the reduction in titer is not due to an excessive number of noninfectious particles with defects in the virus attachment or entry.

## Glycoprotein E2 cell-surface expression is reduced in the absence of 6K and is restored by an efficient ion channel chimera

To determine the role of an ion channel in alphavirus glycoprotein transport to the plasma membrane, we performed IF analysis using antibodies against E2 and CP on BHK-15 cells infected with WT SINV, Δ6K SINV, M2 SINV, Vpu SINV, and P7 SINV with an MOI of 1 at 6 or 12 hpi (Fig 2A). E2 was detected at the plasma membrane at 6 hpi in cells infected with WT SINV with increased accumulation at 12 hpi. In the absence of 6K, E2 was not detected at the plasma membrane at 6 hpi. Instead, E2 accumulated in the ER membrane localizing to the perinuclear region (Fig 2A). Vpu SINV and M2 SINV exhibited E2 expression at the plasma membrane at 6 hpi, with Vpu SINV exhibiting WT-like E2 distribution (Fig 2A). To further validate the E2 expression at the plasma membrane, we used permeabilized and non-permeabilized BHK-15 cells infected with WT and mutant SINV in IF analysis. We quantified the cell-surface expression of E2 using IF analysis of permeabilized and non-permeabilized cells using an anti-E2 antibody (Fig 2B and 2D). The analysis supported the data obtained from the IF analysis depicting the low surface expression of E2 in Δ6K SINV. All three chimeric viruses showed increased E2 surface expression compared to Δ6K SINV, with Vpu SINV the most efficient. We next quantified the cell surface expression of E2 in WT and mutant SINV at 6 and 12 hpi with flow cytometry, using an anti-E2 antibody (Figs 2C and S2) Comparable to the results obtained from the IF analysis, Δ6K SINV showed defective E2 transport to the cell surface. The plasma membrane expression of E2 in Vpu SINV was equivalent to WT SINV.

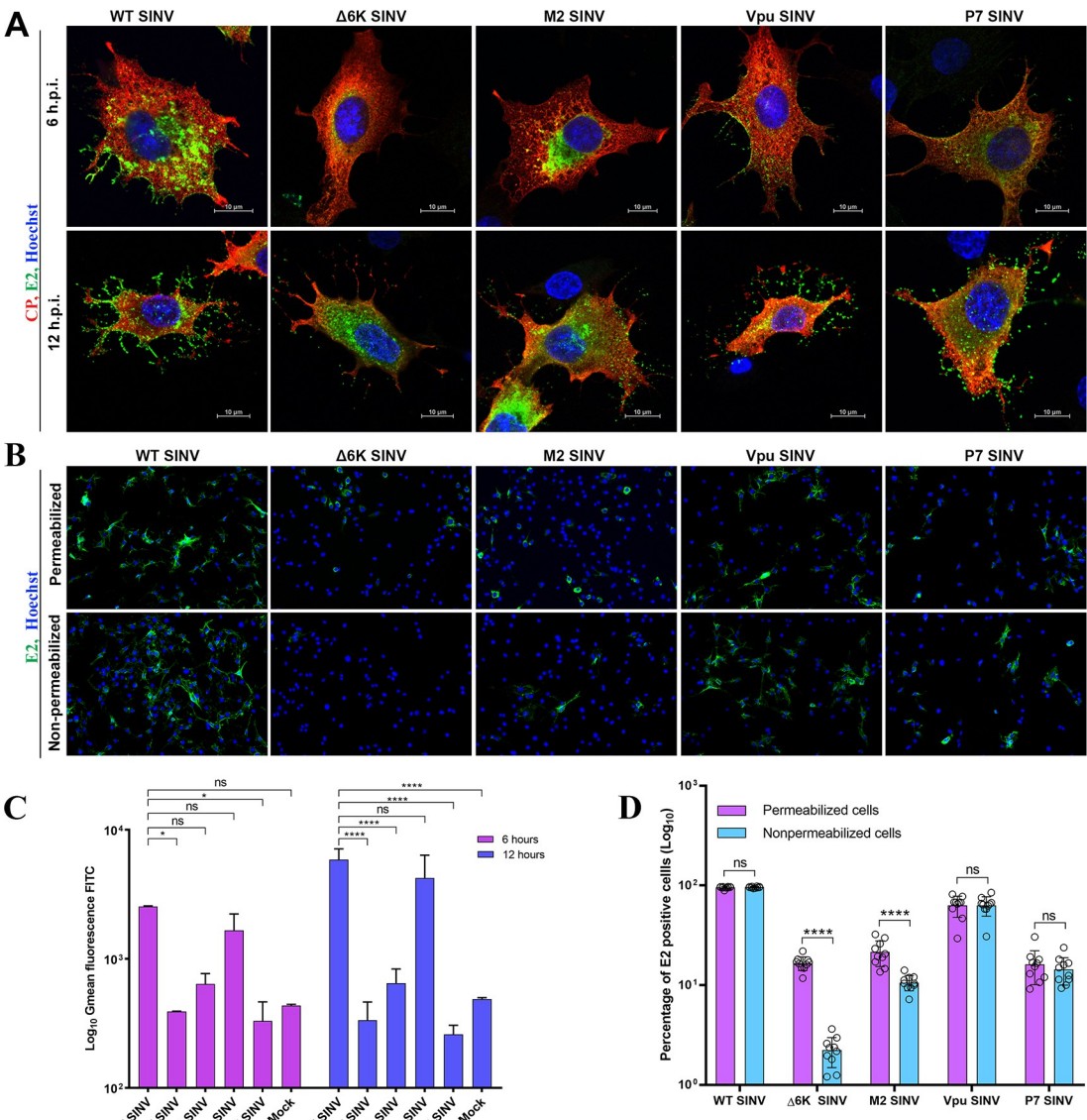

**Fig 2. Functional complementation of 6K by other viral ion channels. (A)** IF analysis of permeabilized BHK-15 cells infected with WT SINV, Δ6K SINV, M2 SINV, Vpu SINV, or P7 SINV at 6 or 12 hpi using antibodies against E2 (Green) and CP (Red). **(B)** IF analysis of glycoprotein trafficking to the plasma membrane under permeabilized or non-permeabilized conditions. BHK-15 cells were electroporated with RNA corresponding to WT or 6K mutant viruses and fixed at 12 h post-electroporation Anti-E2 monoclonal antibody was used to detect glycoproteins. Nuclei were stained with Hoechst stain. **(C)** Flow cytometry analysis of cells infected with WT or 6K mutant viruses at an MOI of 5. Cells were incubated with a monoclonal anti-E2 antibody followed by staining with FITC secondary antibody. Results are shown as a $\log_{10}$ of the geometric mean of the region of the curve measuring the amount of E2 fluorescence at the plasma membrane. Data shown are from two independent experiments. Error bars indicate standard error of mean (SEM). Significance was determined using Dunnett's multiple comparisons test as part of two-way ANOVA with a 95% confidence interval. *p* Values were considered significant when $p < 0.05$ (*), $p < 0.01$ (**), $p < 0.001$(***), and $p < 0.0001$(****). ns indicates "not significant". **(D)** Quantification of glycoprotein trafficking to the plasma membrane from (B). The number of cells expressing E2 (green) relative to the total number of cells (blue) was determined using the NIS Elements software. Cells from 10 independent frames per condition were counted. Error bars indicate standard error of mean (SEM). Significance was determined by multiple unpaired t-tests of data. *p* Values were considered significant when $p < 0.05$ (*), $p < 0.01$ (**), $p < 0.001$(***), and $p < 0.0001$(****). ns indicates "not significant".

Together, these results show that alphavirus glycoprotein transport to the cell surface is restricted in the absence of an ion channel activity, and substituting a functional ion channel enhances the E2 surface expression.

## Live-cell imaging of E2 trafficking to the plasma membrane indicates that in the absence of 6K, E2 accumulates on internal membranes of the secretory pathway

As a first step to examine the alphavirus spike transport in live cells in the absence of virus replication and assembly, we co-expressed a YFP-tagged plasma membrane marker with a SINV envelope glycoprotein construct containing an mCherry tag on E2 (E3-mCherry-E2-6K/TF-E1) or a Δ6K mutant (E3-mCherry-E2-E1) (Fig 3A and 3B). Whereas mCherry-E2 localized to the plasma membrane (Fig 3A and S1 Video) in the WT construct, E2 accumulated on large cytoplasmic vesicles in the Δ6K mutant (Fig 3B and S2 Video). Since E2 cell surface expression occurs as an E2/E1 complex in the spike form, these results indicate that 6K exerts its functional role in alphavirus spike transport independent of virus replication, nonstructural proteins, and capsid. To corroborate these results in the context of live virus, we analyzed the cell-surface expression levels of E2 in live cells infected with mCherry-E2 tagged WT or Δ6K SINV. Cells were electroporated with RNA corresponding to these mCherry-E2 expressing SINV and Δ6K SINV and imaged at 6, 8, and 12 h post-electroporation (Fig 3D). Confirming the IF results and glycoprotein expression of WT and Δ6K mutant SINV, in live-cell imaging, mCherry-E2 was detected on filopodial extensions on the plasma membrane at 6 h post-electroporation of mCherry-E2 SINV, whereas mCherry-E2 accumulated in the perinuclear region in mCherry-E2 Δ6K SINV. Comparably, C6/36 cells infected with the mCherry-E2 tagged WT, or Δ6K SINV exhibited E2 accumulation on the cell surface, and E2 accumulated on large cytoplasmic vesicles in the absence of 6K at 24 hpi (S3 Fig). To verify whether the trafficking defect of E2 in Δ6K SINV is due to glycoprotein processing, we performed a western blot analysis of cell lysates using an anti-E2 antibody (Fig 3C). We observed the processing of E3 and E2 in both WT and Δ6K SINV, although there was a delay in the furin cleavage of E3-E2 unsurprisingly due to the delayed glycoprotein trafficking in Δ6K mutant as this cleavage occurs in late Golgi. Thus, it is evident that the delay in glycoprotein trafficking to the plasma membrane is not due to a polyprotein processing defect in Δ6K SINV.

## Spatio-temporal analysis of 6K exhibits its localization to the ER, secretory vesicles, and the Golgi apparatus

To visualize the spatiotemporal organization of 6K and E2 in the secretory pathway, we investigated the subcellular localization of fluorescently tagged 6K and E2 using live-cell imaging. As a first step to identifying 6K localization, we constructed a fluorescent SINV expressing a miniSOG tag at the N-terminus of 6K (Fig 4A). The miniSOG-6K SINV resulted in a medium plaque phenotype (Fig 1B and 1C) with a 3-log reduction in virus titer compared to WT SINV (Fig 1D), similar to what was previously reported for mCherry tagged-E2 SINV [15,59] and several other fluorescent protein-tagged alphavirus mutants [66–69]. Western blot analysis using an anti-E2 antibody confirmed that although the insertion of the miniSOG-tag causes a delay in glycoprotein processing, a characteristic of alphavirus with fluorescent-protein tags in the structural proteins, the tag did not inhibit E3-E2 cleavage (Fig 4B) suggesting that the miniSOG-6K SINV is suitable for live imaging. Furthermore, the insertion of miniSOG-6K tag was stably retained in SINV for five passages. Thus, we performed live imaging of cells transiently transfected with the ER marker mCherry-SEC61B-C1 (Fig 4C), the trans-Golgi network (TGN) marker mTagBFP2-SiT-N-15 (Fig 4D), or the envelope glycoproteins

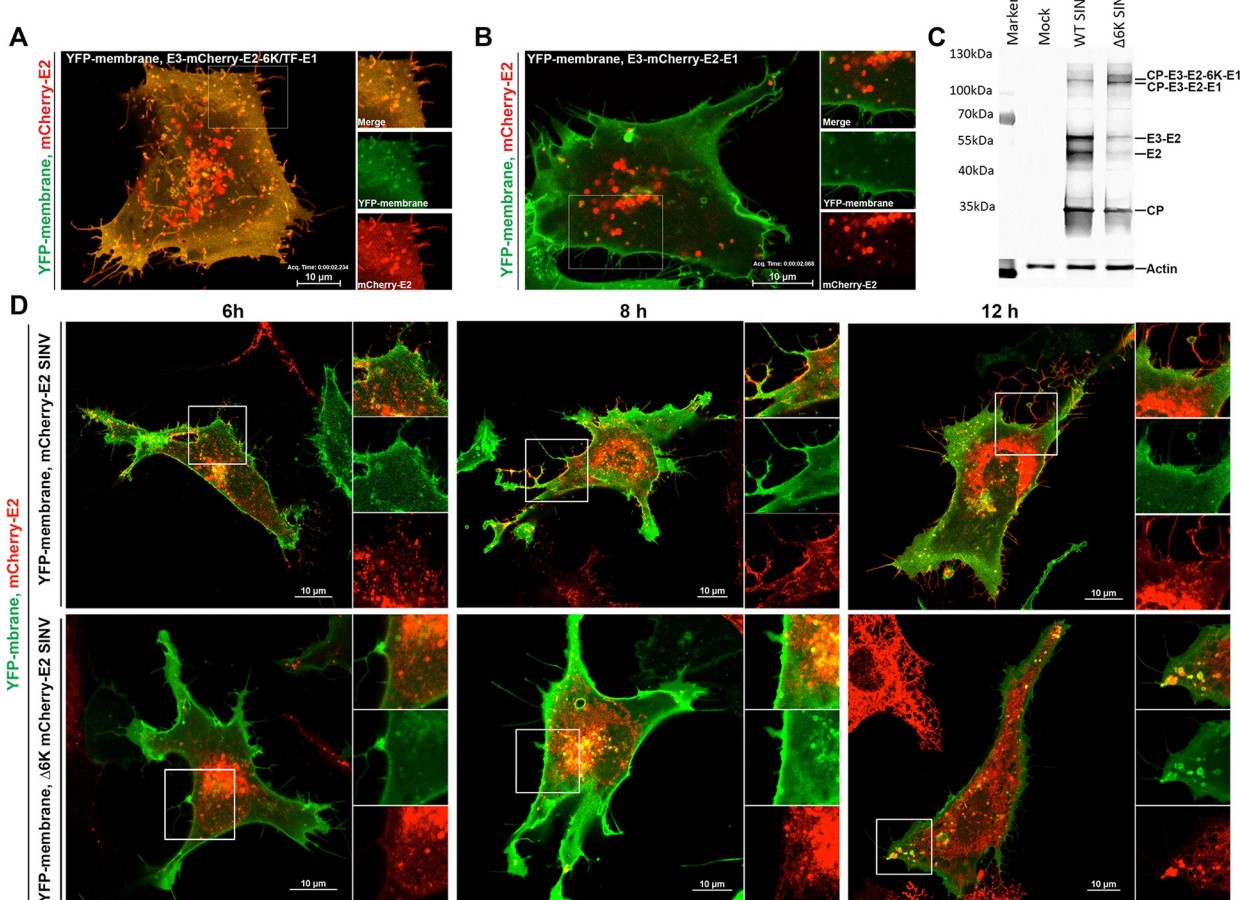

**Fig 3. 6K is required for glycoprotein trafficking to the plasma membrane. (A-B).** Representative images of BHK-15 cells co-transfected with YFP-membrane (green) and E3-mCherry-E2-6K/TF-E1 (red) (A) or E3-mCherry-E2-E1 (red) (B). Images were collected at 12 h post-transfection. Images correspond to videos (S1 and S2 Videos). **(C)** Western blot analysis of cell lysates from BHK-15 cells infected with WT SINV or Δ6K SINV. The blot was probed with an anti-E2 primary antibody to detect the structural polyprotein-processing intermediates. **(D)** Images of BHK-15 cells transfected with YFP-membrane (green) and RNA corresponding to mCherry-E2 SINV (red) or Δ6K mCherry-E2 SINV (red). Images were collected at the indicated time points post-electroporation. Regions of interest are marked and represented as zoomed to the right in separate channels.

CP-E3-mCherry-E2-6K/TF-E1 (Fig 4E), and infected with miniSOG-6K SINV at 12 hpi. From the image analysis, miniSOG-6K was detected localizing predominantly to the ER (S3 Video) and post-Golgi vesicles (S4 Video) colocalizing with E2 in the secretory pathway. However, we did not detect any quantifiable colocalization of miniSOG-6K and E2 on the plasma membrane (S5 Video). To further evaluate the localization of 6K and E2 when processed from the same structural polyprotein, we generated a dual-labeled mCherry-E2/miniSOG-6K SINV. Live-imaging of cells electroporated with RNA corresponding to the dual-labeled virus confirmed the colocalization of 6K and E2 in the secretory pathway (Fig 4F and S6 Video). Furthermore, this colocalization was unaffected by the deletion of TF (Fig 4G and S7 Video), suggesting that beyond the ER, 6K is transported along with E2 to the Golgi apparatus. Similar results confirming the colocalization of 6K and E2 were obtained from C6/36 cells infected with the WT or the ΔTF dual-labeled virus and imaged at 24 hpi (S3 Fig). We next tested whether the localization of 6K and the ion channel chimeras could be confirmed by IF analysis using an epitope tag. To investigate the localization of 6K and the ion channel chimeras, we

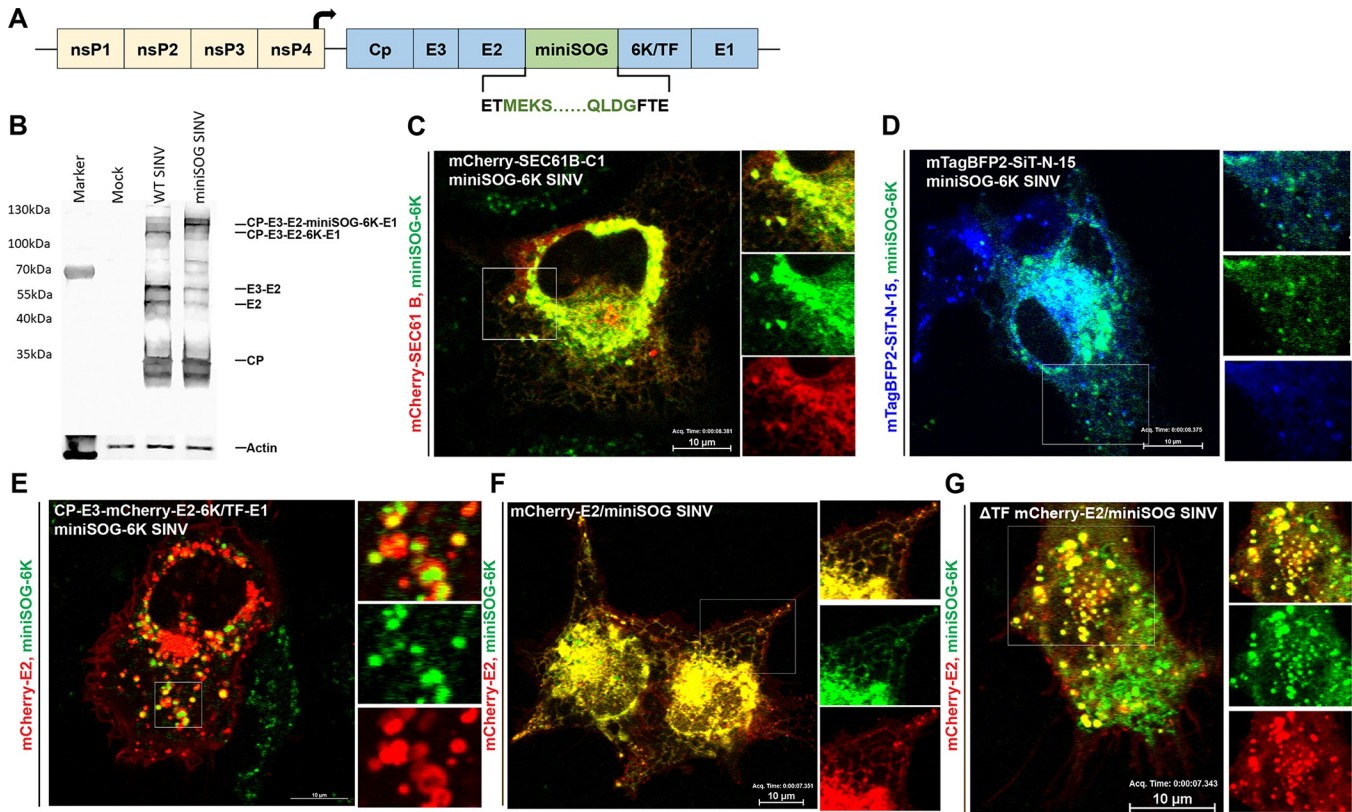

**Fig 4. Subcellular localization of 6K in infected cells. (A)** Schematic of the miniSOG-6K SINV construct. The sequence encoding the fluorescent protein miniSOG was cloned into WT toto64 after $Thr_2$ of 6K as an N-terminal fusion. **(B)** Western blot analysis of the cell lysates from BHK-15 cells infected with WT SINV or miniSOG-6K SINV. The blot was probed with an anti-E2 primary antibody to detect the structural polyprotein processing intermediates. **(C-E)** Representative images of BHK-15 cells transfected with mCherry-SEC61B-C1 (red) (C), mTagBFP2-SiT-N-15 (blue) (D), or CP-E3-mCherry-E2-6K/TF-E1 (red) (E), and infected with miniSOG-6K SINV (green). Imaging was performed at 12 hpi. Regions of interest are marked and enlarged to the right as separate channels. Images correspond to videos (S3–S5 Videos) **(F-G)** Representative images of BHK-15 cells electroporated with RNA corresponding to the dual-labeled mCherry-E2/miniSOG SINV (mCherry-E2 is red and miniSOG-6K is green) (F) or ΔTF mCherry-E2/miniSOG SINV (mCherry-E2 is red and miniSOG-6K is green) (G) and imaged at 12 hpi Images correspond to videos (S6 and S7 Videos).

inserted a Flag tag at the N-termini of 6K, Vpu, M2, and P7 SINV. The addition of a Flag tag to WT SINV, M2 SINV, Vpu SINV, or P7 SINV did not significantly affect virus titer, quantified by plaque assays (S4 Fig). We performed IF analysis on Flag-tagged 6K, Vpu, M2, and P7 SINV to determine the subcellular localization of 6K and ion channel chimeras in infected cells. Due to the availability of tools, we used anti-Giantin antibody and anti-Flag antibody in human hepatoma cells to determine the localization of 6K and other ion channels. Our analysis revealed that all the ion channel chimeras localized to the ER (S4 Fig); however, only 6K (Fig 5A), M2 (Fig 5C), and Vpu (Fig 5B) localize to the Golgi apparatus. The P7 chimera did not localize to Golgi and was retained in the ER (Figs 5D and S4). Furthermore, the localization of Flag-6K in the IF analysis was akin to the miniSOG-6K localization detected by live imaging, confirming that 6K, Vpu, and M2 channels localize to the Golgi apparatus.

## Ion channel activity of 6K is critical for the biogenesis of virus-induced CPV-II structures

To determine the functional role of ion channel activity in the virus-infected cells, we performed TEM analyses of WT and mutant SINV infected cells at 12 hpi (Fig 6). Given the

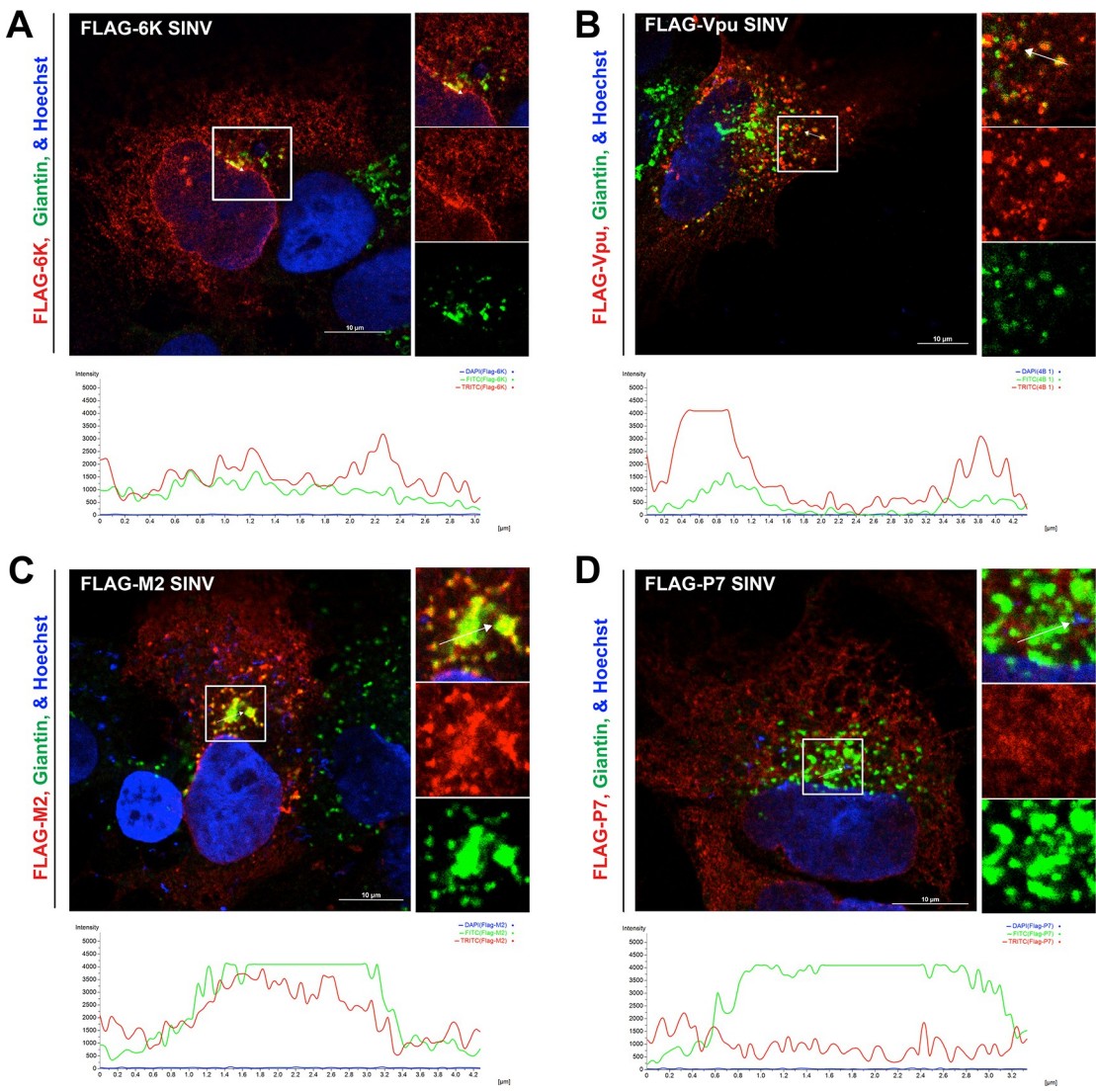

**Fig 5. Localization of 6K and the ion channel chimeras. (A-D)** IF analysis using anti-FLAG (red) and anti-Giantin (green) antibodies of permeabilized huh-7.5 cells infected with an MOI of 5 of (A) FLAG-6K SINV, (B) FLAG-Vpu SINV, (C) FLAG-M2 SINV, or (D) FLAG-P7 SINV. Cells were fixed at 12 hpi. Regions of interest are marked and enlarged to the right as separate channels. Fluorescent profiles of the lined areas are displayed at the bottom. Distance is in μm and intensity is in arbitrary fluorescence units (AFU).

functional role of 6K in the secretory pathway, we hypothesized that 6K might be involved in the formation of virus-induced structures, particularly originating from the Golgi apparatus. We detected replication structures CPV-Is and budding viruses in all the tested samples (Figs 6 and S5). The type II cytopathic vacuoles CPV-IIs with single- and double-membrane were observed in cells infected with WT, Vpu SINV, and M2 SINV (Fig 6A, 6C and 6D). However, we did not detect CPV-II structures in cells infected with Δ6K SINV or P7 SINV in multiple images screened (n = 60) (Fig 6B and 6E). We also confirmed that the absence of CPV-IIs was not due to the lack of TF, as none of the ion channel chimeras is producing TF proteins. Together, our TEM analyses reveal a unique molecular role of 6K in the biogenesis of CPV-IIs in SINV infected cells, which correlated with a functional ion channel in the Golgi apparatus.

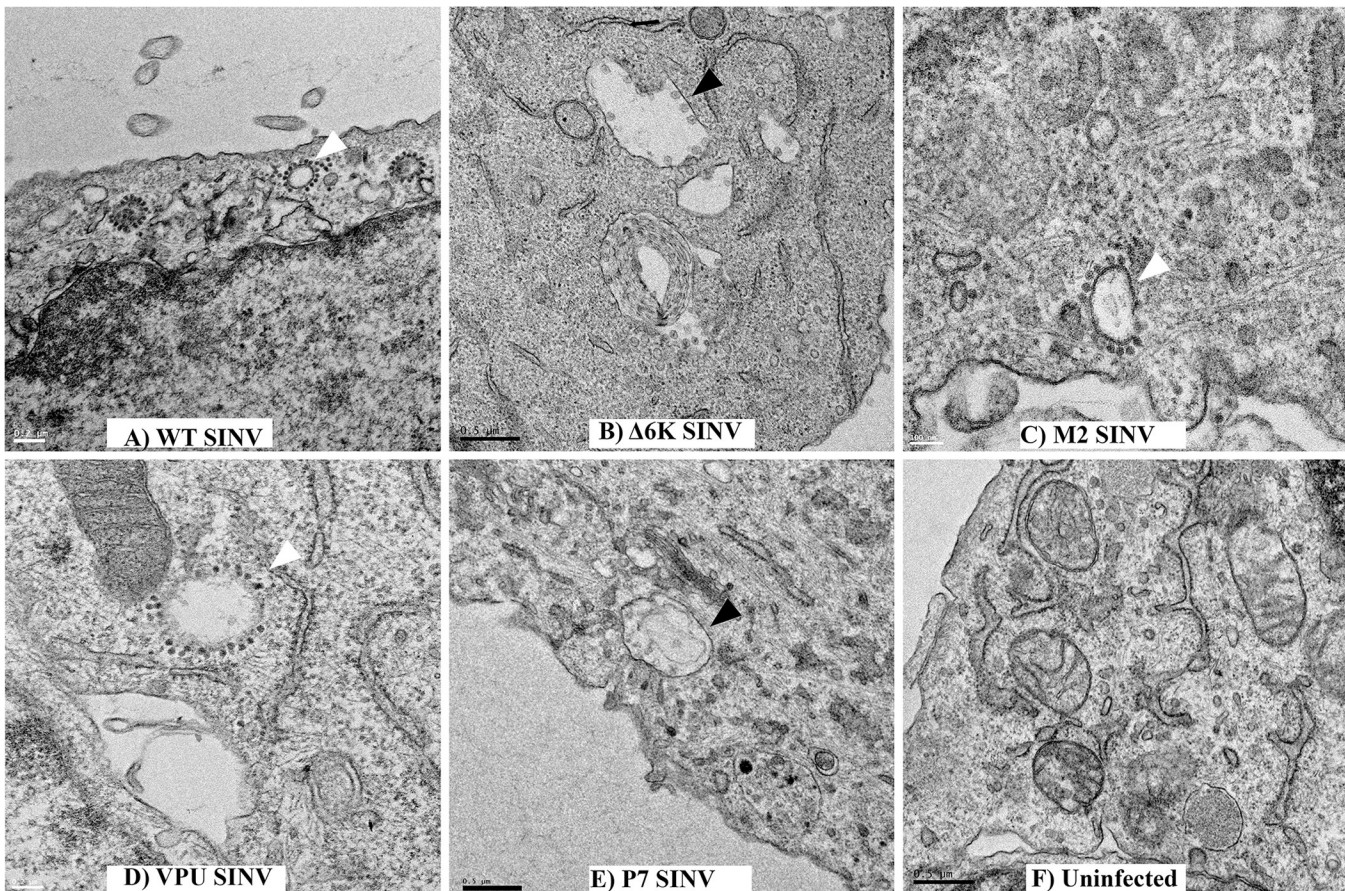

**Fig 6. Ion channel activity of 6K is required for the biogenesis of CPV-IIs.** TEM analysis of BHK-15 cells. Cells were infected with WT SINV (**A**), Δ6K SINV (**B**), M2 SINV (**C**), Vpu SINV (**D**), or P7 SINV (**E**), and fixed at 12 hpi. Uninfected cells (**F**) were used as a control. Black arrowheads indicate CPV-Is, and white arrowheads indicate CPV-IIs. The scale bars are shown in each panel of the figure.

## Inhibition of 6K and ion channel chimeras using channel blocking drugs

Given that the ion channel activity is required for alphavirus glycoprotein trafficking and virus release, we next investigated whether inhibiting the ion channel activity using established channel blockers will reduce virus titers from WT and ion channel chimeras. Since P7 was not localizing to the Golgi, we proceeded with M2 and Vpu chimeras for the channel blocking experiments. We tested amantadine and hydroxy methylene amiloride (HMA), channel blockers inhibiting influenza M2 and HIV Vpu, respectively. Cytotoxicity of these drugs at different concentrations was determined using alamarBLUE (Fig 7A and 7B). Our observations presented here suggest that amantadine reduces M2 SINV virus titer by 1-log at 0.5 mM concentration, whereas at the same concentration, amantadine treatment resulted in less than a log reduction in WT and Vpu SINV titers (Fig 7C). HMA effectively reduced the WT SINV and Vpu SINV in a concentration-dependent manner, resulting in a 2-log reduction for WT SINV and a 1.2-log reduction for Vpu SINV at a 40 µM concentration. On the contrary, M2 SINV was not significantly inhibited by HMA treatment resulting only in a less than a half-log reduction when treated with a 40 µM of the drug (Fig 7D). This demonstrates a strict correlation between the type of ion channel and their specific inhibitor in the SINV chimera and that 6K is functionally analogous to the ion channel Vpu than M2 suggesting 6K as a potential antiviral target.

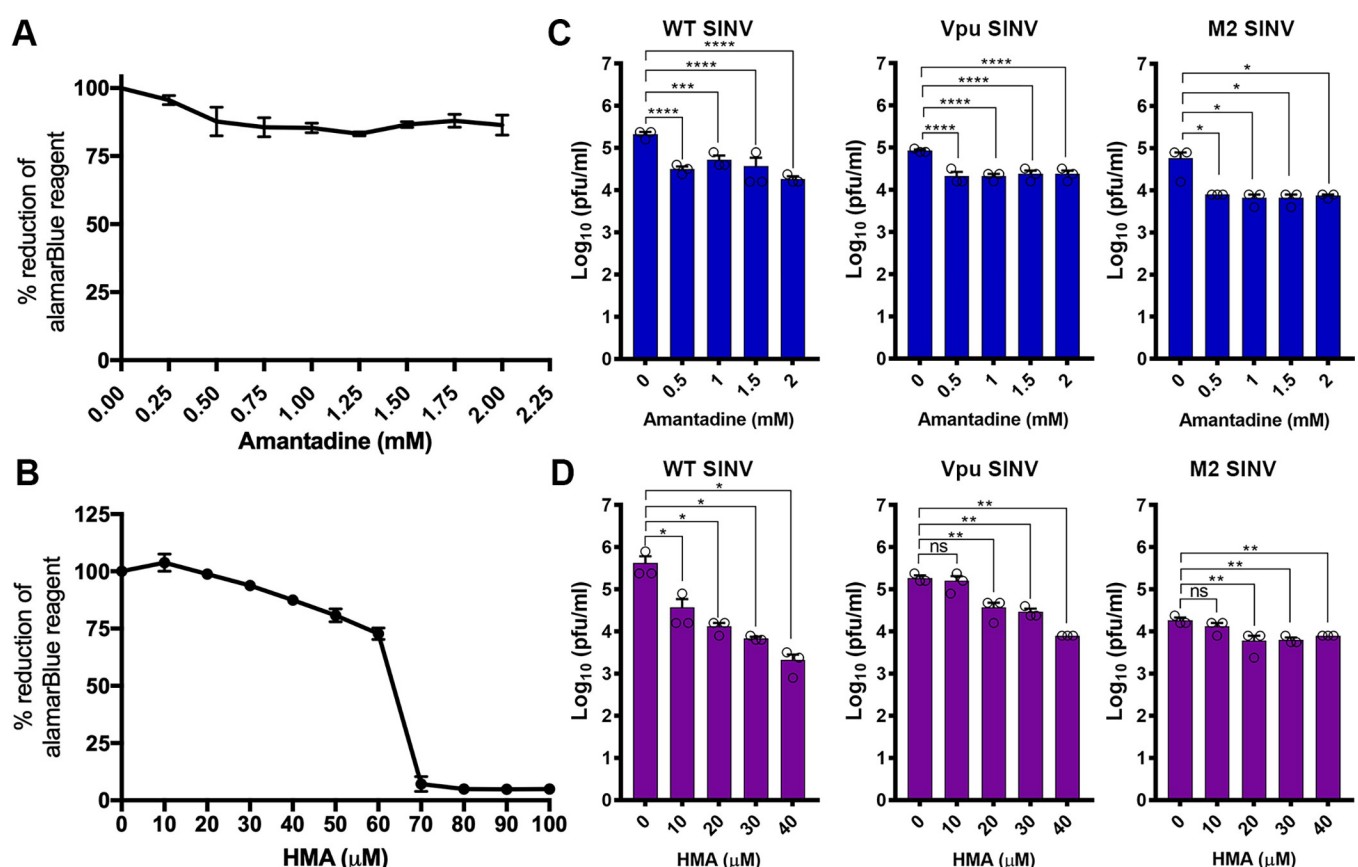

**Fig 7. Effect of HMA and amantadine on virus release from SINV and ion channel chimeras. (A-B).** Cytotoxicity of Amantadine (A) and HMA (B) was calculated using alamarBLUE reagent **(C-D).** BHK-15 cells were pretreated with the corresponding concentrations of HMA and Amantadine for 24 h and then infected with WT or 6K mutant viruses at an MOI of 0.1. Viral titers were determined using the supernatants collected at 12 hpi. Data shown are from three independent experiments. Error bars indicate standard error of mean (SEM). Significance was determined using Dunnett's multiple comparisons test as part of one-way ANOVA with a 95% confidence interval. *p* Values were considered significant when p < 0.05 (*), p < 0.01 (**), p < 0.001(***), and p < 0.0001(****). ns indicates "not significant".

## Discussion

Viral ion channels contribute to different stages in the virus life cycle, including entry, genome replication, and assembly; however, their primary role is to participate in virion morphogenesis and release from infected cells [42]. The structure and function of integral oligomeric bundles formed by the ion channels encoded by several clinically important enveloped viruses, including influenza A virus matrix protein 2 (M2) [70], hepatitis C virus p7 [71], HIV-1 viral protein U (Vpu) [72], and SARS-CoV-2 envelope protein E (E) [73] have been characterized. Alphavirus 6K has been shown to have weak ion selectivity and the ability to induce leakage of solutes into and out of the cell across a sealed membrane similar to the hexameric ion channel of HCV p7 [71] and the pentameric channel of SARS-CoV-2 envelope small membrane protein (E) [73]. Although 6K has been shown to modulate the intracellular membrane permeability and interact with spike proteins to promote spike maturation, the three-dimensional structure and the mechanism by which it imparts a significant role in the alphavirus life cycle remain unknown [74]. Furthermore, the TF protein produced by a (−1) programmed ribosomal frameshifting in a heptanucleotide slip site in the 6K coding region is incorporated into the virion [37,75,76]. Although TF has been shown to be incorporated into purified SINV

particles [30], its structural organization in the virion has not been resolved even in the 3.5 Å cryo-EM structure of SINV [24], presumably due to its low stoichiometric packaging [32,77].

To characterize the mechanism by which 6K exerts its function on the alphavirus lifecycle, we generated a mutant SINV, Δ6K SINV, containing an in-frame deletion of 6K and TF (Fig 1A). Comparable to what has been reported for RRV and SFV, the absence of 6K negatively affected the production of SINV (Fig 1D) [31,78]. Previously, using pulse-chase experiments and surface biotinylation, SFV 6K has been shown to be dispensable for the heterodimerization of the pE2 (E3-E2) and E1, spike membrane protein transport to the cell surface, and virus production [79]. By contrast, we quantitatively show that the Δ6K SINV is defective in glycoprotein transport to the plasma membrane in BHK-15 and C6/36 cells using live-imaging and IF analyses (Figs 2 and S3). These spike glycoproteins that are defective in transport to the cell surface also accumulate on large cytoplasmic vesicles in the perinuclear regions in the Δ6K SINV infected cells. Thus, our glycoprotein trafficking results agree with previous reports that showed impaired glycoprotein trafficking in two SINV 6K mutants when 15 amino acids were inserted at position 29 of or 24–45 amino acids were deleted [36,80]. As previously observed for 6K deletions in other alphaviruses, our observations presented here suggest that the furin cleavage of pE2 into E3 and E2 was not inhibited in Δ6K SINV (Fig 3C). However, the cleavage is delayed presumably due to defects in glycoprotein trafficking to the Trans-Golgi network where the furin cleavage occurs [81,82]. These results suggest that the delay in glycoprotein trafficking to the plasma membrane in Δ6K SINV is not due to the defects in envelope glycoprotein processing but due to the lack of an essential function carried out by 6K in the secretory pathway.

The ion channel activity of bacterially expressed 6K and recombinant 6K have been studied previously. 6K has been shown to form cation-selective ion channels when inserted into planar lipid bilayers [38,39], by a conserved transmembrane helix (S1 Fig) which is shared between both 6K and TF [32]. To investigate whether the defects caused by the absence of 6K are due to the lack of a functional ion channel, we generated a P7 SINV chimera expressing full-length P7 in place of 6K. We chose P7 since the membrane topologies of 6K, and P7 are similar, with their N and C termini oriented towards the ER lumen [83,84]. Although P7 SINV showed a 10-fold higher titer compared to Δ6K SINV, it formed small plaques, and virus titers were ~1000 fold reduced compared to the WT virus, suggesting that HCV P7 is not adequate to replace 6K functionally (Fig 1D). We next substituted the ion channel TM helix of 6K with the TM helix of HIV Vpu and Inf M2. Unlike 6K, the C-termini of both M2 and Vpu orient to the cytoplasmic side. Although more efficient than P7 SINV, the M2 SINV was not as effective in rescuing the defects in virus growth kinetics and titer compared to Vpu SINV. Comparing the results from all the viral ion channels tested indicates that Vpu is the most efficient in restoring WT-like growth kinetics (Fig 1D). As evidenced from the flow cytometry and IF analyses, M2 and P7 are not as efficient as Vpu in complementing glycoprotein trafficking defects observed in the absence of 6K. A full-length Vpu expressed in trans under a double subgenomic promoter in SINV has previously been shown to partly rescue the defects caused by a partial deletion of 6K [36]. Corroborating these findings, our results also demonstrate that the Vpu ion channel TM helix alone in cis is sufficient to rescue glycoprotein transport to the plasma membrane and SINV release, also indicating that the ion channel activity of 6K is functionally analogous to Vpu. Interestingly, recent biochemical and computational studies proposed a novel topological model for the alphavirus structural polyprotein in which 6K has a single pass transmembrane helix similar to Vpu [85,86]. The inability of Vpu to completely restore virus budding could be due to the absence of TF from the chimeric virus as previous studies showed that TF is essential for SINV infectious particle release [37]. Additionally, the absence of 6K

could also be affecting virus budding since studies showed that 6K mediates E2-CP interactions during alphavirus budding [87].

Given that a functional ion channel activity is critical for promoting SINV release, we pursued to determine the subcellular localization of 6K in live cells during virus infection. Previously, IF analyses of RRV infected cells [78] and studies using 6K expression constructs have shown that 6K mainly localizes to the ER [37,88]. Here, we show the colocalization of Flag-tagged 6K with the Golgi marker Giantin. Both Vpu and M2 displayed similar colocalization to Giantin, whereas P7 did not colocalize with Giantin, and was primarily localized to ER (Figs 5 and S4). The ER retention of P7 possibly explains its deficiency in functional complementation compared to Vpu and M2. Subsequently, we inserted a pH tolerant small fluorescent mini-iSOG tag at the N-terminus of 6K for real-time imaging [54]. Utilizing the miniSOG-6K SINV along with cellular markers in real-time live-cell imaging we identified that in addition to being present in the ER, 6K traffics to the late secretory pathway localizing to dynamic vesicles (Fig 5). To evaluate the colocalization of 6K and E2 in the secretory pathway specifically on the dynamic vesicles that contain 6K, we generated a dually labeled mCherry-E2 and miniSOG-6K SINV. Upon infection of BHK cells, we found that 6K colocalizes with E2 on ER, Golgi, and highly dynamic vesicles in the secretory pathway using live imaging (Fig 5). CHIKV 6K protein expression studies have shown the colocalization of 6K and E2 in the secretory pathway and the trafficking of 6K is influenced by the presence of E2 [88].

We next tested whether the localization of Flag 6K and miniSOG 6K in the TGN is exclusively due to the presence of TF protein which has been reported to be packaged into virions. We mutated the ribosome slip site from the mCherry-E2/miniSOG-6K SINV to stop the production of TF protein. 6K localized to E2-positive post-Golgi vesicles in the absence of TF, indicating that 6K traffics to the TGN along with E2 (Fig 5). Furthermore, we detected the localization of 6K along with E2 on the limiting membranes of large E2-positive vacuoles in C6/36 cells infected with the dual-labeled SINV (S3 Fig). These static vacuoles that are intermediates of CPV-Is and CPV-IIs observed in mammalian cells are also the site of virus budding in mosquito cells [15]. MiniSOG-6K was also detected in smaller E2-positive dynamic vesicles presumably containing internally budded virions trafficking to the plasma membrane. Although we have not tested TF localization in the absence of 6K, our findings do not exclude the possibility that TF can also be localizing to post-Golgi vesicles since both TF and 6K were detected on the plasma membranes of cells infected with SFV [32]. Despite this very compelling evidence for localization of 6K on the internal membranes in the secretory pathway, our live-imaging data did not reveal an accumulation of 6K on the plasma membrane. This discrepancy could be due to the possibility that 6K is present in very few copies at substoichiometric level compared to E2 at the plasma membrane [29].

We set out to identify the existence of virus-induced structures and sites of virus budding in BHK cells infected with WT and mutant SINV using TEM. Whereas CPV-Is and budding viruses were detected in all samples analyzed, CPV-IIs were not detected in some of the mutant viruses. Vpu SINV, and M2 SINV showed the presence of CPV-II. However, these structures were not found in cells infected with Δ6K SINV and P7 SINV (Fig 6). The absence of CPV-II observed for Δ6K SINV, and P7 SINV demonstrates a strict correlation between CPV-II formation and the presence of an ion channel in the TGN. In the case of P7, this could be due to its restricted localization to the ER membrane where the replication and assembly of HCV occur [89,90]. Furthermore, it may be because p7 activity is regulated by its specific interaction with HCV non-structural protein 2 (NS2), which is absent in our system [91]. Additionally, unlike P7, 6K traffics to the TGN from the ER membrane and reaches the late secretory pathway, where it has a functional role in inducing the production of CPV-IIs. [89,92]. Since CPV-IIs originate from the medial/trans-Golgi [19], Golgi fragmentation may contribute to

the formation of CPV-IIs [20]. Virus-induced Golgi fragmentation was previously reported for positive-sense RNA viruses such as picornaviruses and coronaviruses [93–95]. Interestingly, in the case of coronaviruses, Golgi fragmentation is induced by the accessory protein open reading frame 3a (ORF3a) which assembles into tetrameric ion channels [96–98]. Additionally, disruption of the pH gradient in the secretory pathway and the increase in intracellular calcium levels can affect the integrity of the Golgi apparatus [99–101]. Collectively, our results demonstrate that the ion channel activity of 6K or the presence of a functional ion channel at the TGN is required for the biogenesis of CPV-IIs.

HCV p7 and IAV M2 have been shown to reduce the acidification of intracellular vesicles and cellular organelles [43,102]. A significant role of M2 is to protect influenza virus envelope protein hemagglutinin (HA) from premature conformational changes as it transits through the low-pH compartments in the exocytic pathway to the plasma membrane [103]. M2 plays a subtle role in regulating the pH of the transport pathway as it can increase the vesicular pH by as much as 0.8 pH units to protect the structural integrity of HA [104]. It has been proposed that the ion channel activity of M2 regulates the pH balance between the acidic lumen of TGN and the pH of the cytoplasm to protect HA from premature low-pH-induced conformational changes in the TGN [103]. Our results strongly support attributing a similar function to 6K, where after the furin cleavage of pE2 in the TGN, the ion channel activity of 6K protects the glycoprotein spikes from a low pH mediated rearrangement of E1, leading to premature fusion in TGN [105,106]. Similarly, Vpu, a multimer in the native state, affects membrane conductance by modulation of endogenous ion channels [107]. Furthermore, HIV-1 Vpu modifies the activity of cellular K+ channel TASK-1 to promote virus release [108]. Thus far, alphavirus 6K has not been shown to regulate the activity of any host ion channels; it remains a possible mechanism of 6K function. However, viroporins can induce efflux of ions, such as H+ and Ca2+, that move from their intracellular stores in the ER and the Golgi apparatus into the cytoplasm following a strong electrochemical gradient, altering the pH and calcium homeostasis in ER and Golgi [109]. Since Ca2+ is required for membrane fusion events, its leakage can cause inhibition of anterograde vesicle trafficking [110]. The viral ion channel enterovirus 2B protein has been shown to reduce the ER and Golgi complex Ca (2+) levels and inhibit protein trafficking through the Golgi complex, presumably by forming transmembrane pores [109]. Therefore, ion channel activity of 6K might lead to reduced Ca(2+) levels in the ER and Golgi complex, causing the inhibition of CPV-II formation and glycoprotein transport as observed in Δ6K SINV.

Amantadine and HMA are effective pharmacological inhibitors that significantly reduce the production of virions by inhibiting the viral ion channel activity. Amantadine, a drug that targets the ion channel activity binds M2 protein at the N-terminal lumen of the channel blocking ion conductance [50,92,111]. Previous studies showed that Amantadine can inhibit the release of CHIKV particles [88]. HMA has been shown to inhibit HIV-1 by binding and blocking the Vpu channel activity [52,112–114]. We hypothesized that the ion channel chimeras would be sensitive to their specific channel inhibitors, blocking the channel activity and reducing virus titers. Therefore, we tested amantadine and HMA in virus inhibition studies. We found that HMA can inhibit virus release from BHK cells infected with WT SINV or the chimeric Vpu SINV, whereas M2 SINV was not significantly inhibited by HMA treatment (Fig 7). Based on the WT-like growth kinetics of Vpu SINV and inhibition of virus release by channel blocking drug HMA, we propose that the oligomeric state of 6K and its ion channel conductance are similar to that of HIV-1 Vpu. Our results also indicate that the oligomeric state of 6K may be different from the tetrameric M2 ion channels structurally a less favorable oligomeric arrangement for 6K.

In this study, we provide the spatial and temporal organization of 6K and attribute the functional significance of the ion channel activity of 6K to virus assembly. We show that efficient

glycoprotein trafficking to the plasma membrane and CPV-II formation are dependent on the presence of a functional ion channel in the TGN. CPV-IIs are also associated with glycoprotein trafficking to the plasma membrane; thus, investigating the functional role of ion channels in the reorganization of the secretory pathway in cells infected with alphaviruses can shed more light on the role of CPV-IIs in alphavirus assembly and budding. We also show that 6K localizes to the TGN, which colocalizes with E2. Our observations suggest that 6K is functionally similar to the HIV-1 Vpu, and its ion channel activity can be inhibited by HMA, an effective drug against Vpu. These findings open new avenues for therapeutic intervention strategies based on rational drug design of 6K ion channel inhibitors. Virus release was only partially rescued by the other ion channels used in this study, indicating that 6K may have other functions, such as interaction with other viral and host proteins. Determining the monomeric and oligomeric structures of 6K will facilitate our understanding of the mechanism of ion channel activity and the other possible functions of this essential protein.

## Supporting information

**S1 Fig. Multiple sequence alignment of 6K protein from alphaviruses.** Alignment files were generated using CLUSTAL omega and sequence alignments were viewed using Jalview.
(JPG)

**S2 Fig. Flow plot of flow cytometry.** Representative flow charts of cells infected with WT or 6K mutant viruses at an MOI of 5. Cells were incubated with a monoclonal anti-E2 antibody followed by staining with FITC secondary antibody.
(JPG)

**S3 Fig. Live-cell imaging of C6/36 cells. (A-B).** Representative images of C6/36 cells infected with mCherry-E2 SINV (red) or Δ6K mCherry-E2 SINV (red) and imaged at 24 hpi. **(C-D).** Representative images of C6/36 cells infected with the dual-labeled mCherry-E2/miniSOG SINV (red and green) (C) or the ΔTF mCherry-E2/miniSOG SINV (red and green) (D) and imaged at 24 hpi.
(JPG)

**S4 Fig. Localization of 6K and the ion channel chimeras.** IF analysis using anti-FLAG antibody of permeabilized BHK-15 cells transfected with mEmerald-SEC61B-C1 and infected with **(A)** FLAG-6K SINV, **(B)** FLAG-Vpu SINV, **(C)** FLAG-M2 SINV, or **(D)** FLAG-P7 SINV. Cells were fixed at 12 hpi. **(E)** Plaque assays of WT and Flag-tagged viruses. Data shown are from three independent experiments. Error bars indicate standard error of mean (SEM). Significance was determined by multiple unpaired t-tests of data. p Values were considered significant when $p < 0.05$ (*), $p < 0.01$ (**), $p < 0.001$(***), and $p < 0.0001$(****). ns indicates "not significant".
(JPG)

**S5 Fig. Virus budding occurs in cells infected with WT or chimeric viruses.** TEM analysis of BHK-15 cells. Cells were infected with WT SINV **(A)**, Δ6K SINV **(B)**, M2 SINV **(C)**, Vpu SINV **(D)**, P7 SINV **(E)**, or miniSOG-6K SINV **(F)**, and fixed at 12 hpi. White arrowheads indicate budding viruses.
(JPG)

**S1 Video. Live-cell imaging of BHK-15 cells co-transfected with YFP-membrane (green) and E3-mCherry-E2-6K/TF-E1 (red).**
(MP4)

**S2 Video.** Live-cell imaging of BHK-15 cells co-transfected with YFP-membrane (green) and E3-mCherry-E2-E1 (red).
(MP4)

**S3 Video.** Live-cell imaging of BHK-15 cells transfected with mCherry-SEC61B-C1 (red) and infected with miniSOG-6K SINV (green). Imaging was performed at 12 hpi.
(MP4)

**S4 Video.** Live-cell imaging of BHK-15 cells transfected with mTagBFP2-SiT-N-15 (blue) and infected with miniSOG-6K SINV (green). Imaging was performed at 12 hpi.
(MP4)

**S5 Video.** Live-cell imaging of BHK-15 cells transfected with CP-E3-mCherry-E2-6K/TF-E1 (red) and infected with miniSOG-6K SINV (green). Imaging was performed at 12 hpi.
(MP4)

**S6 Video.** Live-cell imaging of BHK-15 cells electroporated with RNA corresponding to the dual-labeled mCherry-E2/miniSOG SINV (red and green) and imaged at 12 hours post-electroporation.
(MP4)

**S7 Video.** Live-cell imaging of BHK-15 cells electroporated with RNA corresponding to the dual-labeled ΔTF mCherry-E2/miniSOG SINV (red and green) and imaged at 12 hours post-electroporation.
(MP4)

**S1 Table. List of primers used for mutagenesis and cloning.**
(DOCX)

## Acknowledgments

We thank Laurie Mueller for TEM experiments and the use of the Bioscience Imaging Facility of the Bindley Bioscience Center and Life Science Microscopy Facility, Purdue University.

## Author Contributions

**Conceptualization:** Zeinab Elmasri, Richard J. Kuhn, Joyce Jose.

**Data curation:** Zeinab Elmasri, Joyce Jose.

**Formal analysis:** Zeinab Elmasri, Vashi Negi, Richard J. Kuhn, Joyce Jose.

**Funding acquisition:** Richard J. Kuhn, Joyce Jose.

**Investigation:** Zeinab Elmasri, Vashi Negi, Richard J. Kuhn, Joyce Jose.

**Methodology:** Zeinab Elmasri, Joyce Jose.

**Project administration:** Richard J. Kuhn, Joyce Jose.

**Resources:** Joyce Jose.

**Supervision:** Richard J. Kuhn, Joyce Jose.

**Validation:** Vashi Negi.

**Visualization:** Zeinab Elmasri, Vashi Negi, Joyce Jose.

**Writing – original draft:** Zeinab Elmasri, Joyce Jose.

**Writing – review & editing:** Zeinab Elmasri, Vashi Negi, Richard J. Kuhn, Joyce Jose.

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
