## [Decision Letter · Decision Letter 0]

21 Jun 2022

Dear Dr. Jose,

Thank you very much for submitting your manuscript "Requirement of a functional ion channel for alphavirus glycoprotein transport, CPV-II formation, and efficient virus budding." for consideration at PLOS Pathogens. As with all papers reviewed by the journal, your manuscript was reviewed by members of the editorial board and by several independent reviewers. In light of the reviews (below this email), we would like to invite the resubmission of a significantly-revised version that takes into account the reviewers' comments.

Elmasri et al find the 6K protein in Sindbis can be replaced by the Vpu protein from HIV, and less so by M2 and p7 from Influenza and HCV, respectively. They go on to propose that 6K and E2 colocalize in the Golgi and these proteins together are responsible for CPV II formation, promoting E2-E1 glycoprotein transport. Finally, they conclude that the ion channel activity of 6K is what is driving CPV II formation and glycoprotein transport. Very little work has been done on 6K and its role as an ion channel, the scientific impact of this research is high. However, as both the reviewers and myself point out, there are some concerns with the conclusions drawn from the results.

The major issues addressed by more than one reviewer:

1. A majority of the conclusions are based on correlation and no evidence of a direct interaction between 6K and E2. Some suggestions include complementation experiments--will expression of E2 and 6K be enough to form CPV II? Do the chimera viruses also form CPV IIs? Any ideas how ionic flux is mediating CPV II formation and why are these vesicle more likely to transport glycoproteins to the plasma membrane? More suggestions are in the reviewer comments. In addition, better images including more quantification, for both IF colocalization and TEM CPV II formation would strengthen the manuscript.

2. Figure 3 is not clear. nor is it complete The text, figure, and legend indicated different constructs being used making drawing conclusons difficult. More analysis on the trafficking of E2 to the plasma membrane is necessary. For example, is E1 produced? does it transit to the plasma membrane? The western blot doesn't show if E1 is produced (side note: The western is with anti-E2 yet Capsid os on the blot).

3. Why is Vpu able to replace 6K so efficiently and M2 and p7 cannot? Some further discussion is warranted, even structural predictions of the different putative ion channel proteins. It should also be addressed that recent work has shown 6K might not be two transmembrane helices but rather a single pass TM helix, similar to Vpu. Does the coronavirus E protein substitute for 6K?

We cannot make any decision about publication until we have seen the revised manuscript and your response to the reviewers' comments. Your revised manuscript is also likely to be sent to reviewers for further evaluation.

Sincerely,

Suchetana Mukhopadhyay

Guest Editor

PLOS Pathogens

Sara Cherry

Section Editor

PLOS Pathogens

Kasturi Haldar

Editor-in-Chief

PLOS Pathogens

orcid.org/0000-0001-5065-158X

Michael Malim

Editor-in-Chief

PLOS Pathogens

orcid.org/0000-0002-7699-2064

Elmasri et al find the 6K protein in Sindbis can be replaced by the Vpu protein from HIV, and less so by M2 and p7 from Influenza and HCV, respectively. They go on to propose that 6K and E2 colocalize in the Golgi and these proteins together are responsible for CPV II formation, promoting E2-E1 glycoprotein transport. Finally, they conclude that the ion channel activity of 6K is what is driving CPV II formation and glycoprotein transport. Very little work has been done on 6K and its role as an ion channel, the scientific impact of this research is high. However, as both the reviewers and myself point out, there are some concerns with the conclusions drawn from the results.

The major issues addressed by more than one reviewer:

1. A majority of the conclusions are based on correlation and no evidence of a direct interaction between 6K and E2. Some suggestions include complementation experiments--will expression of E2 and 6K be enough to form CPV II? Do the chimera viruses also form CPV IIs? Any ideas how ionic flux is mediating CPV II formation and why are these vesicle more likely to transport glycoproteins to the plasma membrane? More suggestions are in the reviewer comments. In addition, better images including more quantification, for both IF colocalization and TEM CPV II formation would strengthen the manuscript.

2. Figure 3 is not clear. nor is it complete The text, figure, and legend indicated different constructs being used making drawing conclusons difficult. More analysis on the trafficking of E2 to the plasma membrane is necessary. For example, is E1 produced? does it transit to the plasma membrane? The western blot doesn't show if E1 is produced (side note: The western is with anti-E2 yet Capsid os on the blot).

3. Why is Vpu able to replace 6K so efficiently and M2 and p7 cannot? Some further discussion is warranted, even structural predictions of the different putative ion channel proteins. It should also be addressed that recent work has shown 6K might not be two transmembrane helices but rather a single pass TM helix, similar to Vpu. Does the coronavirus E protein substitute for 6K?

Reviewer's Responses to Questions

**Part I - Summary**

Reviewer #1: In this study, Elmarsi et al., look to understand the function of the Sindbis virus (SINV) 6K ion channel. They generate a SINV 6K deletion as well as chimeric viruses with known viral ion channels (very cool!). They show that 6K is required for virion production and glycoprotein trafficking to the plasma membrane. In addition, they found that Vpu can rescue virion production and E2 trafficking. Using live cell imaging, they show that the SINV E2 accumulates in the secretory pathway when 6K is absent, and that 6K localizes to the secretory pathway. TEM experiments show differences in CPV structures. Finally, they show that SINV is sensitive to the known Vpu inhibitor HMA suggesting that 6K is similar to Vpu and an antiviral target.

Together, this is a well-written study that provides new insight into SINV biology and begins to elucidate the role of 6K in SINV infection. However, there are several concerns regarding the study including experiments and the conclusions drawn.

Reviewer #2: The manuscript by Elmasri et al. analyzes a prominent member of the viroporin family – the 6K protein from alphaviruses. While the importance of 6K in the life cycle of alphaviruses is somewhat well established, the exact requirement for the ion channel activity is not understood. The authors have generated several chimeras by combining or replacing 6K in SINV by viroporins from other families like HCV p7, Influenza M2 and HIV Vpu, and tested their effect on virus particle production and infectivity, glycoprotein trafficking and other cellular phenomenon like formation of Type II CPVs. The comments and concerns regarding this work are as follows:

Major points:

1) The main findings of the study are a correlation between the availability of an effective ion channel activity, and the formation of type-II cytopathic vacuoles, which in turn correlates with spatiotemporally correct glycoprotein trafficking and virus particle release. The primary results are correlation based, and no direct interactions/effects are shown in the manuscript, which is a major flaw. For example, is there any direct interaction between the glycoprotein E2 and 6K/6K substitutes during infection or under in vitro conditions? This appears a very important point to establish the role of 6K in promoting glycoprotein trafficking.

2) Also, what about the transport of E1? Do the constructs used in the study - E3-mCherry-E2-E1 and E3-mCherry-E2-6K/TF-E1 - have a cleavage site before E1? If so, the live cell imaging panel cannot ascertain where E1 is ending up since the tag is on E2.

3) The authors find that Vpu closely mimics the functionality of 6K, followed by M2, and finally p7, which does not appear to be functionally close to 6K even though its membrane topology is closest to that of 6K. This anamoly is not probed further. In terms of sequence or predicted secondary or tertiary structural elements, is 6K closer to Vpu/M2 compared to p7?

4) Does overexpression of 6K or 6K+E2 in cells result in the formation of CPV-II, or does it only happen during SINV infection – this may point to the involvement of other viral proteins/RNA in the process?

5) The manuscript mentions the differences in virus titre and specific infectivity of virus particles generated upon 6K deletion or 6K replacement with chimeric components. It will be great if this information is provided on the same scale (log vs fold decrease). A 2.8 fold decrease in specific infectivity of particles without 6K suggests that the defect is not only in release, but also in the assembled particles themselves. In this context, it is important to morphologically characterize SINV particles without 6K, or those containing 6K chimeras, in comparison to WT particles. This will add important information to the study.

6) Finally, there appears to be some difference in the behavior of alphaviruses with 6K deleted, as well as the susceptibility of 6K from different alphaviruses to drugs/inhibitors. How similar are the alphavirus 6K proteins in terms of sequence or structure?

7) Along the same lines, it will be helpful to include the established roles of M2/Vpu/p7 in the life cycles of the respective viruses in the Introduction/Discussion for purposes of comparison to 6K.

Minor point:

The manuscript includes substantial experimental details in the Discussion section. The Results and Discussion sections should be reorganized for clarity.

Reviewer #3: The manuscript: Requirement of a functional ion channel for alphavirus glycoprotein transport, CPV-II formation, and efficient virus budding, but Elmasri, Negi, Kuhn, and Jose, uses a variety of imaging and chimeric virus methodologies to provide an in depth characterization of the functional role of the viral 6K protein's ion pore activity in virion maturation and budding. While the pore forming activity of the alphavirus 6K protein, and its potential as an antiviral target has been well recognized within the alphavirus field, the specific stages in the viral lifecycle where ion channel activity acts have not been well characterized. Therefore, the study is important, and in general, the manuscript is well written, the experiments are well done, and the authors' conclusions are supported by the data. However, as noted below, there are some concerns that should be addressed to provide further support for the authors' conclusions, clarify discrepancies in the manuscript, or make it easier for the reader to follow.

**Part II – Major Issues: Key Experiments Required for Acceptance**

Reviewer #1: 1. A major concern is that, as the authors state throughout the paper, functional studies of 6K have been completed in SFV. The authors have nicely compared their work to these previous works and shown that SINV is different than SFV. This finding is very cool and a comparative analysis of SFV in this paper would have been interesting. Nonetheless, as written, this paper is SINV specific and the authors should replace “alphavirus” throughout the paper, including the title, with SINV.

2. Figure 1D shows a 1-step growth curve yet the y-axis is a “rate of virus release”. Can the authors also show the actual growth curve titers over time in the figure? I think it would be much easier to interpret this way. From the current figure, even the 6K deletion is pumping out 10K viruses/ml/hr. That seems like a lot still. Moreover, when comparing Figure 1 to Figure 7, WT, Vpu, and M2 viruses grow similarly in Figure 7 at MOI 0.1 (no drug) yet there are very different in Figure 1 (MOI = 5). Can the authors comment on these differences?

3. Figure 3. A few points: A) The text states that these experiments are in the absence of capsid (Line 368-369) yet the figure and figure legend states that capsid is present. Can you confirm which is correct? These are important for your conclusion on line 373.

B) It is hard to see differences in Figure 3C. Can you use the membrane marker to make this clearer? Also, are there less filopodia in the delta6K virus? Could this be a phenotype?

C) The protein processing is interesting. However, to conclude that the phenotype is not due to processing, I think showing complementation would strengthen your argument. Can you express 6K in trans (as you can with Vpu) and show that processing is restored? Or use the Vpu chimera virus?

4. All of your BHK experiments are completed in 12 hours. Could timing be an issue for the 6K mutant? Throughout the manuscript you use the word “Delayed” which implies time. If you waited longer, do you get more processing of the proteins in Figure 3D?

5. Figure 4: Here you use two tagged viruses that seem to behave very differently. In Figure 4E, the 6K and E2 are punctate and then in Figure 4F 6K and E2 are diffuse and seem to be on filopodia. While I don’t doubt 6K is in the secretory pathway, you may have different results for E2 (different compartments) depending on the system? Can you use the dual tagged virus with ER or golgi markers? Also, what does the protein processing or virion production look like in the dual tagged virus? Is it restored?

6. The conclusion about CPV formation is interesting but I think more work needs to be done on this section. The TEM images are hard to see and interpret. I suggest zooming in and using some of your trans systems to prove your point. Regardless, this needs significant work for this submission.

7. Your inhibitor experiment is awesome and an amazing tool to dissect the function of 6K in the absence of a deletion (which could have bigger issues). I think looking at protein processing, E2 trafficking, and CPV formation in the presence of HMA would add significantly to your conclusions.

Reviewer #2: 1) Analysis of direct binding of 6K to E2, under in vitro conditions or during co-expression in cells/infection

2) Same for 6K and E1

3) Formation of CPV-II in cells during overexpression of 6K/co-expression of 6K with viral proteins

4) Bioinformatics based comparative study of viroporins used in the manuscript

5) Morphological analysis of SINV particles with 6K deleted/6K chimeras

Reviewer #3: 1) If ion channel activity is the driver of CPV-II formation, HMA (or amantadine) treatment would be expected to inhibit CPV-II, but not CPV-I formation. This should be tested and the data included in the study, or if the results are negative, discussed.

2) The electron microscope data from Fig. 6 should be quantified (e.g. total number of CPV-I and CPV-II per number of fields) for each virus/mutant.

3) Given that several of the constructs show delays in processing, the authors may want to soften some of their language ruling out any effect of processing kinetics on their phenotypes, or more clearly lay out why this is not an issue.

**Part III – Minor Issues: Editorial and Data Presentation Modifications**

Reviewer #1: (No Response)

Reviewer #2: A thorough discussion of the sequential/structural differences between 6K proteins from different alphaviruses

Comparison of the roles of M2/Vpu/p7 in the respective viral life cycles

Reorganisation of the Discussion section

Reviewer #3: a) The authors should consider moving Supplemental Table 2 into Figure 1 to make the data more accessible to the reader.

b) For Fig. 2A, please clarify which fluorophore (green vs red) is used for E2 vs CP to make is easier for non-expert readers to follow.

c) For Fig. 2c, representative flow plots should be included as supplemental data.

d) Please check the label on the image for Fig. 3B for accuracy. Based on the text it should be CP-E3-mCherryE2-E1, but on the image it is CP-E3-mCherryE2-6K/TF-E1.

e) When discussing ion channel inhibitors as antivirals, please include discussion of reference 77 (Dey, et al), since that study evaluated amantadine's effects on chikungunya virus.

PLOS authors have the option to publish the peer review history of their article (what does this mean?). If published, this will include your full peer review and any attached files.

Reviewer #1: No

Reviewer #2: No

Reviewer #3: No
---

## [Editor Report · Decision Letter 1]

21 Sep 2022

Dear Dr. Jose,

We are pleased to inform you that your manuscript 'Requirement of a functional ion channel for Sindbis virus glycoprotein transport, CPV-II formation, and efficient virus budding' has been provisionally accepted for publication in PLOS Pathogens.

Best regards,

Suchetana Mukhopadhyay

Guest Editor

PLOS Pathogens

Sara Cherry

Section Editor

PLOS Pathogens

Kasturi Haldar

Editor-in-Chief

PLOS Pathogens

orcid.org/0000-0001-5065-158X

Michael Malim

Editor-in-Chief

PLOS Pathogens

orcid.org/0000-0002-7699-2064

The authors have sufficiently addressed the points raised by the reviewers.
---

## [Editor Report · Acceptance letter]

29 Sep 2022

Dear Dr. Jose,

We are delighted to inform you that your manuscript, "Requirement of a functional ion channel for Sindbis virus glycoprotein transport, CPV-II formation, and efficient virus budding," has been formally accepted for publication in PLOS Pathogens.

Best regards,

Kasturi Haldar

Editor-in-Chief

PLOS Pathogens

orcid.org/0000-0001-5065-158X

Michael Malim

Editor-in-Chief

PLOS Pathogens

orcid.org/0000-0002-7699-2064